# Rapid fast-delta decay following prolonged wakefulness marks a phase of wake-inertia in NREM sleep

Jeffrey Hubbard [1✉], Thomas C. Gent [2,3], Marieke M. B. Hoekstra[1], Yann Emmenegger[1], Valerie Mongrain [4], Hans-Peter Landolt [5,6], Antoine R. Adamantidis[2,7] & Paul Franken [1✉]

Sleep-wake driven changes in non-rapid-eye-movement sleep (NREM) sleep (NREMS) EEG delta (δ-)power are widely used as proxy for a sleep homeostatic process. Here, we noted frequency increases in δ-waves in sleep-deprived mice, prompting us to re-evaluate how slow-wave characteristics relate to prior sleep-wake history. We identified two classes of δ-waves; one responding to sleep deprivation with high initial power and fast, discontinuous decay during recovery sleep (δ2) and another unrelated to time-spent-awake with slow, linear decay (δ1). Reanalysis of previously published datasets demonstrates that δ-band heterogeneity after sleep deprivation is also present in human subjects. Similar to sleep deprivation, silencing of centromedial thalamus neurons boosted subsequent δ2-waves, specifically. δ2-dynamics paralleled that of temperature, muscle tone, heart rate, and neuronal ON-/OFF-state lengths, all reverting to characteristic NREMS levels within the first recovery hour. Thus, prolonged waking seems to necessitate a physiological recalibration before typical NREMS can be reinstated.

[1] Center for Integrative Genomics, University of Lausanne, Lausanne, Switzerland. [2] Department of Neurology, Inselspital University Hospital Bern, Bern, Switzerland. [3] Department of Veterinary Anesthesia, University of Zürich, Zürich, Switzerland. [4] Department of Neuroscience, Université de Montréal, Montreal, QC, Canada. [5] Institute of Pharmacology and Toxicology, University of Zürich, Zurich, Switzerland. [6] Sleep & Health Zürich, University of Zürich, Zürich, Switzerland. [7] Department of Biomedical Research, University of Bern, Bern, Switzerland. ✉email: jeffrey.hubbard@unil.ch; paul.franken@unil.ch

Research on sleep function is based on the premise that this behavioral state remedies the wear and tear caused by preceding waking. To gain insights into the underlying neuronal substrates, researchers focused on variables that accumulate during waking and dissipate across sleep, reflecting a process known as sleep homeostasis. Nap and sleep deprivation (SD) studies in birds and mammals demonstrated that EEG delta (~0.75–4.5 Hz; δ-)power during non-rapid-eye-movement sleep (NREMS) is in a quantitative relationship with prior sleep-wake history, wherein δ-power increases during early NREMS proportionally with increasing wake duration[1–4]. Moreover, modeling approaches established that changes in δ-power can be accurately predicted by the sleep-wake distribution[2,5–9].

This predictability of EEG δ-power as a sleep-need marker has led to its widespread use in the field. Moreover, several influential hypotheses on sleep regulation and function are based on how fluctuations in this variable reflect homeostatic sleep need[10–13]. δ-power is quantified through frequency-domain analysis[14] and merges amplitude and incidence information of slow-waves (SWs) into a single metric. Time-domain analysis, quantifying other SW aspects, has yielded new insights into their functional significance[15–17]. For example, changes in SW-slopes and their waveform profiles, are believed to reflect neuronal connection recalibration that strengthens during preceding waking[12,18–20].

SWs are the defining electrophysiological feature of NREMS, with a frequency-range comprising both the slow-oscillation (SO; <1 Hz) and δ-oscillations[21,22], the latter being especially prevalent during NREMS after extended periods of wakefulness[14,23]. Here we use the term SW to refer to both types of oscillations. It is still unclear how the various aspects of SWs respond to sleep-wake history and what their neuro-anatomical substrates and functions are. Through detailed analysis of what causes observable SW-period shortening following prolonged periods of waking, we find that the well-known δ-power changes are in fact a composite of the contributions of two separate SW-populations each with profoundly different sleep-wake driven dynamics. Specifically, only SWs of the faster population are sensitive to prior time-spent-awake and returned to baseline levels much quicker than slower ones, findings which are similarly observed in humans. Furthermore, optogenetic silencing of the centromedial thalamus (CMT) in mice affected only activity of this faster SW-population, indicating that besides a dynamic distinction, there are anatomical differences in their origin. Finally, the dynamics of faster SWs are paralleled by that of several physiological variables in both species not thought to reflect sleep homeostasis. This study identifies a previously unknown complexity of central and peripheral processes associated with the aftereffects of prolonged waking, which we refer to as wake-inertia[24], exemplified by the short-lived decay of a specific sub-population of SWs, before reverting to levels typical of NREMS. This presents an inherent paradox, as the deepest levels of NREMS, when recovery is presumed to be most efficient, in fact display several signatures more reminiscent of waking.

## Results

**Slow-waves accelerate after prolonged waking periods.** We first examined the sleep-wake driven dynamics of frequency-domain (i.e. δ-power; 0.75–4.0 Hz) and time-domain (the SW properties slope, amplitude, and period; Supplementary Fig. 1) features of SWs during NREMS. In a first dataset, 38 male C57BL/6J mice were recorded for 4 continuous days including baseline, a 6-h SD, and recovery. Under baseline conditions, NREMS showed a time-course typical of laboratory mice with high stable levels during the 12-h light periods, lowest in the first 6 h of the dark period, and intermediate levels in the last 6 h (Fig. 1e). The SD resulted in

NREMS increases during subsequent recovery, especially in the dark period. Consistent with the fact that sleep-wake history drives changes in δ-power[2,9], the highest levels were observed following prolonged periods of waking (in the baseline dark period and immediately after SD), whilst during times when NREMS prevails (the baseline light period and recovery after SD), δ-power gradually decreased to stable low levels (Fig. 1a). Time-course analyses of SW-slope and -amplitude, revealed similar sleep-wake driven dynamics as δ-power, yet their range was smaller (Fig. 1b, c). Interestingly, we found that SW-period was shortest immediately following long periods of either spontaneous or enforced waking (Fig. 1d), suggesting that SWs accelerated when sleep homeostatic drive is high.

**δ-band heterogeneity in sleep-wake driven dynamics.** To gain insight into which SW-features contribute to this short-lasting shortening of SW-period we analyzed the first 6 h of SD recovery in more detail by focusing on periods with the largest changes in SW-period (the first 8 of the 25 NREMS quantiles into which the first 6 h of recovery was divided; ≈first 2 h). The highest initial amplitude and quickest subsequent decreases were for SWs with periods between 0.25 and 0.5 s (i.e., 2.0–4.0 Hz; Fig. 2a, Supplementary Fig. 2a). In addition, the incidence of these faster SWs was affected the most at this time (Supplementary Fig. 2b). Importantly, we discovered a bi-modality in SW-prevalence with peaks centered at 1.25 and 3.0 Hz (Fig. 2b). This bimodality in SW-prevalence was maintained throughout the experiment (Supplementary Fig. 3) indicating that their separation does not depend on sleep pressure. Correlations between SW-amplitude and -slope by frequency in the first 8 recovery quantiles where also highest ($R > 0.9$) above 2.0 Hz (Fig. 2c). Thus >2.0 Hz SWs not only contributed the most to increases in overall SW-amplitude observed after SD, but also to that of SW-slope (Fig. 1b, c). Frequency-domain analyses confirmed this frequency-specificity of recovery dynamics (Supplementary Fig. 2g).

We then devised several approaches to further characterize the heterogeneity in the recovery dynamics within the δ-band. First, we estimated the time-point during recovery at which each 0.25 Hz bin had lost half of its initial power (Supplementary Fig. 2e) and found that faster δ-band frequencies (2.0–4.5 Hz) showed steeper declines compared to slower ones (0.5–1.75 Hz). Analyses of EEG spectral recovery dynamics confirmed that the highest initial levels were reached in these faster oscillations (Fig. 2d) and found the largest decreases to occur between 2.5 and 4.5 Hz, while, surprisingly, power in the slowest frequencies (0.5–1.0 Hz) initially increased (Fig. 2e). Finally, unbiased hierarchical clustering of EEG power density time-courses at 0.25 Hz resolution during these first 25 NREMS quantiles, separated a lower-frequency (0.5–2.75 Hz) from a higher-frequency (2.5–3.5 Hz) δ-band (Supplementary Fig. 2c). The aggregate results of these approaches (Fig. 2e, Supplementary Fig. 2c, e, f) and the SW-incidence distribution (Fig. 2b), strongly indicated two distinct δ sub-bands separated at ~2.0–2.25 Hz. As the demarcation of these two bands did not appear static, we chose five representative 0.25 Hz bins for each (δ1: 0.75–1.75 Hz; δ2: 2.5–3.5 Hz).

We then reanalyzed the SD recovery dynamics for these two δ-bands separately, within individual consolidated NREMS episodes. The initial values (first quantile) of δ2 were considerably higher than δ1 (261 ± 13 vs. 196 ± 5%, paired t-test, $t_{37} = -4.6$, $p = 5.1E-5$). Furthermore, recovery dynamics during the first 6 h differed between the two, with δ1, surprisingly, initially increasing and subsequently decreasing linearly, and δ2 falling rapidly within the first hour before abruptly changing to a more gradual

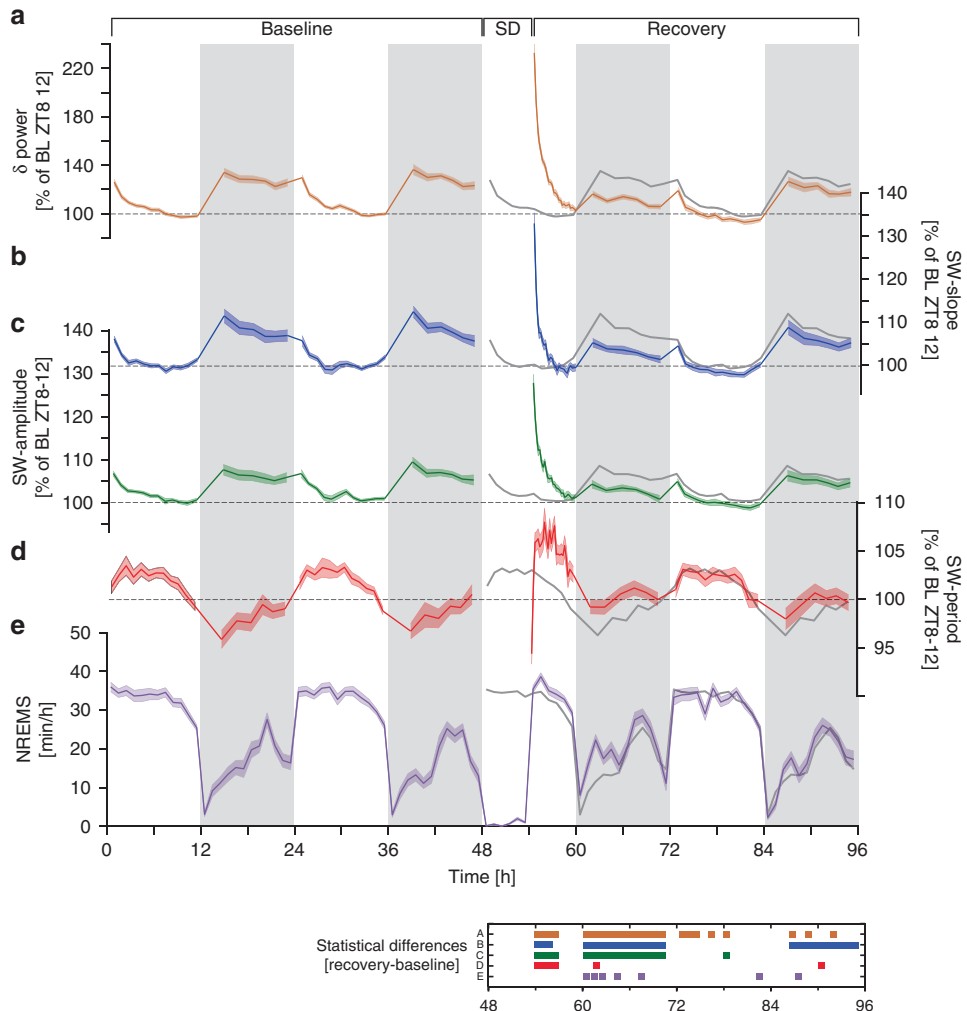

**Fig. 1 Frequency- and time-domain analysis of SWs before and after sleep deprivation. a** Relative changes in δ-power (0.75–4 Hz) followed sleep-wake distribution (see **e**) and were highest after sleep deprivation (SD; 48–54 h; two-way rANOVA factors SD x time: $F_{34,1836} = 115.3$, $p < 1.0E-17$). **b** SW-slope and **c** SW-amplitude followed nearly identical patterns (two-way rANOVA factors SD x time; SW-slope: $F_{34,1819} = 45.2$, $p < 1.0E-17$; SW-amplitude: $F_{34,1819} = 61.6$, $p < 1.0E-17$) although their dynamic range was smaller because power scales as a square to amplitude. **d** Dynamics of SW-period differed from the other three variables and was at times in opposition (two-way rANOVA factors SD × time: $F_{34,1819} = 8.3$, $p < 1.0E-17$). Data in **a–d** represent quantiles [light (12), dark (6), and recovery day 1 light (25)], containing equal numbers of NREMS 4-s epochs, respectively, and were referenced to their respective lowest values reached in baseline (two-day mean of last 4 h of light period). **e** NREMS time-course expressed as min h$^{-1}$ of recording. NREMS increased during recovery compared to baseline (two-way rANOVA factors SD × time: $F_{34,1836} = 3.1$, $p = 1.1E-08$). All values represent mean ± s.e.m (shading); **a**, **e** $n = 38$, **b–d** $n = 37$. Averaged baseline values (Baseline days 1–2) are re-plotted during recovery for comparison (dark-gray line). Significant post-hoc recovery-baseline differences ($p < 0.05$) are presented in a color-matched panel (**a–e** refer to figure panels) below. Note that SW-power, -slope, and -amplitude all fall below baseline during the 1st recovery dark phase. Data are derived from fronto-parietal bipolar EEG signals.

decline (Fig. 2f, Supplementary Fig. 4a). To estimate the time-point where the rate of δ2-decrease changed (pivot-point), dynamics were subjected to two-segment piecewise linear regression (Fig. 2f). A pivot-point could be identified in all mice at 59.7 ± 7.2 min after NREMS onset. The decay in δ1 did not differ before and after the δ2 pivot-point (paired t-test, $p = 0.55$) and was much slower than δ2 before pivot-point (δ1: −35 ± 15; δ2: −166 ± 12%h$^{-1}$; t-test, $t_{37} = 6.86$, $p = 1.0E-5$), but did not differ after (δ1: −26 ± 4; δ2: −20 ± 3%h$^{-1}$; paired t-test, $t_{37} = -1.27$, $p = 0.24$). We compared fits provided by piecewise linear regression against an exponential decay function, generally used to describe decreases in δ-power. Although both functions significantly fit these data, residuals associated with piecewise regression were significantly smaller (Supplementary Fig. 4b, c). As pivot-points and decay-rates varied across animals, averaging the recovery time-courses of δ2-power nevertheless appeared to

follow an exponential decreasing function (Supplementary Fig. 6b).

**δ2-waves are nested inside δ1-waves.** We then reanalyzed waveform characteristics of δ1- and δ2-waves during initial NREMS episodes after SD. δ1-waves (Fig. 2g) resembled what has been previously described in rodents and humans as multi-peak SWs[19,25,26]. Conversely, δ2-wave profiles were uniformly sinusoidal (Fig. 2g). These waveforms persisted when the raw signal was analyzed and did not depend on EEG derivation (Supplementary Fig. 2d; see insert). To further explore δ1-wave composition, raw signals were filtered along a narrower band encompassing only δ2-frequencies (2.0–4.5 Hz). Surprisingly, we found that after SD the vast majority of δ1-waves contained at least one nested δ2-wave (84 ± 2%) and, of those, almost two-thirds contained at least two (65 ± 2%), representing 15 ± 1% of

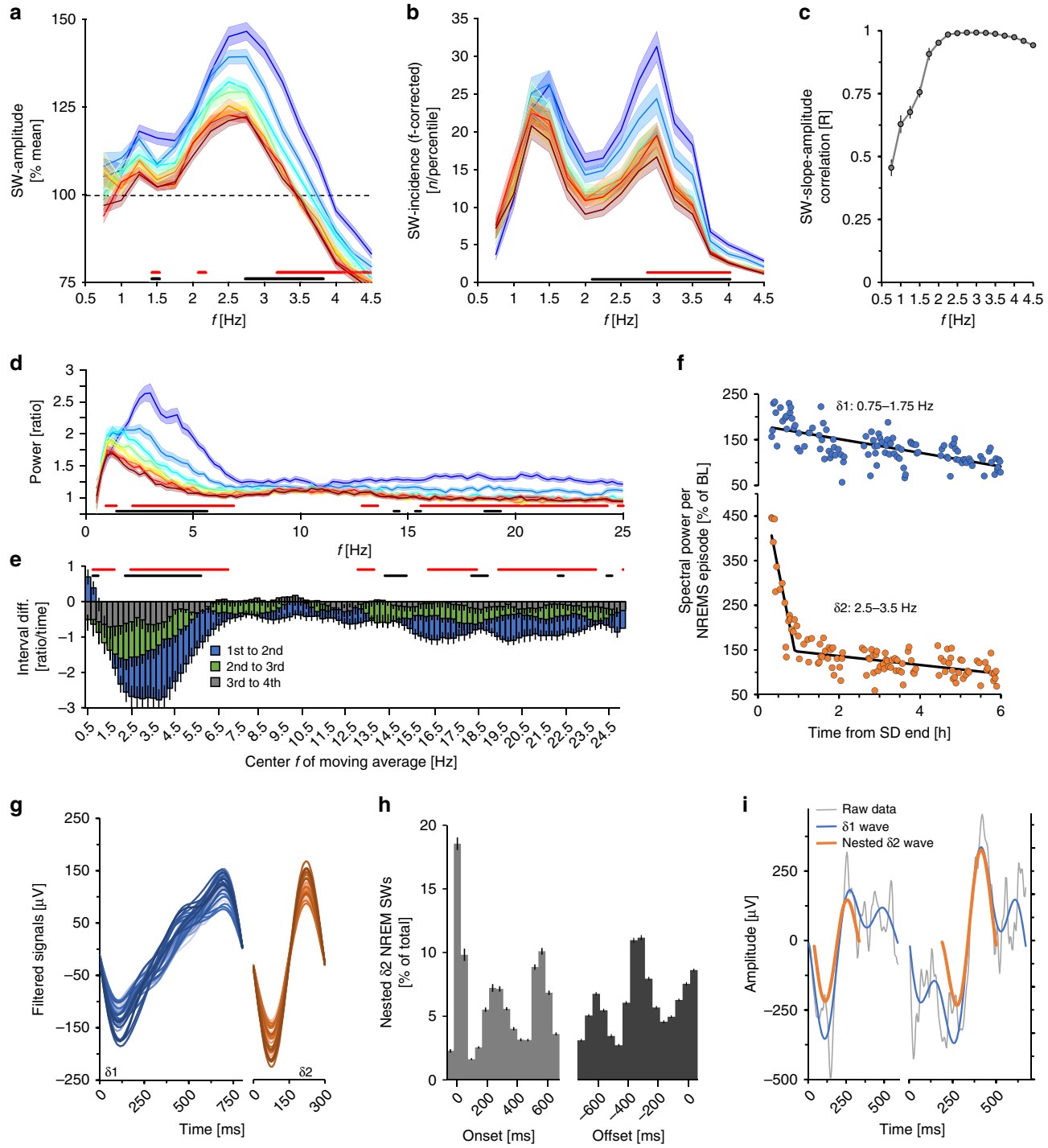

all δ2-waves across the first 6 h of recovery. Nested δ2-waves occurred predominantly at the start or end of a δ1-wave, and at times in its center (Fig. 2h, i).

Nesting was more prevalent in NREMS after periods of prolonged wakefulness, especially after SD and quickly reverted to the lowest baseline levels within the first 6 h of recovery sleep (Supplementary Fig. 5d, e). The percentage of all δ2-waves that were nested followed, however, a different time-course (Supplementary Fig. 5c). Since δ2-wave prevalence, both in absolute number (Fig. 2b) and expressed per minute of NREMS (Supplementary Fig. 5b), was initially much higher than that of δ1-waves, which additionally increased before decreasing

(Supplementary Fig. 4a), the fraction of nested δ2-waves out of all δ2-waves was low immediately after prolonged waking periods (Supplementary Fig. 5c).

**δ2-power, but not δ1-power reflects prior sleep-wake history.** Given the radical differences in the two δ-bands immediately following SD, we probed their dynamics under various conditions. At wake-to-NREMS transitions immediately at recovery onset (i.e., in the 1st of the 25 NREMS quantiles; $8.9 \pm 0.5$ min NREMS), δ2-power increased more rapidly compared to δ1, reaching maximum levels around 32 s (Fig. 3a). Rise-to-maximum regressions indicated a >5-fold faster buildup for δ2

**Fig. 2 Evidence for two distinct δ-bands.** Distribution of **a** SW-amplitude and **b** SW-incidence according to SW-period (frequency; 0.25 Hz bins) during recovery (REC). The 1st 6 h of recovery were divided into 25 NREMS quantiles (7.0 ± 0.1 min NREMS/quantile), the dynamics of the 1st eight are depicted (≈ first 2 h). SW-incidence was corrected for frequency (n $Hz^{-1}$) because slower SWs cannot physically occur as often as faster SWs. **c** SW-slope to -amplitude correlations during the same time were highest in faster frequencies. **d** Relative spectral power during the 1st 2 h of recovery, displays not only a significant decrease in power, but a shift from faster to slower frequencies. Significant differences between the 1st/2nd (two-way rANOVA factors quantile × frequency: $F_{98,7252} = 8.65$, $p < 1.0E{-}17$) and 2nd/3rd quantiles ($F_{98,7252} = 2.08$, $p = 3.24E{-}09$), though not after 3rd/4th ($F_{98,7252} = 1.13$, $p = 0.18$). **e** Ratios of five 0.25 Hz-bin moving averages between spectra in the 1st–2nd, 2nd–3rd, and 3rd–4th time quantiles in **d**. Note that lower frequencies initially increase (two-way rANOVA factors quantile difference × frequency: $F_{99,7326} = 9.13$, $p < 1.0E{-}17$). **f** Time-course of δ1- and δ2-power during the 1st 6 h of recovery per NREMS episode in one representative mouse. Symbols represent mean power reached in consecutive episodes. **g** Average waveforms of detected δ1- (blue) and δ2 (orange)-waves during the 1st 10 min of recovery NREMS following SD, for individual mice (n = 37). **h** Timing of nested δ2-waves relative to δ1-onset (left) or -offset (right; 0 ms indicates onset and offset, respectively). Histograms span 50 ms time-bins plotted at midpoint. Of note, the 2nd and 3rd histogram peaks indicate the starts of these additional nested δ2-waves (n = 37; one-way rANOVA factor time-bin; δ2-onset: $F_{15,540} = 305.2$, $p < 1.0E{-}17$; δ2-offset: $F_{15,540} = 327.4$, $p < 1.0E{-}17$). **i** Example of a δ2-wave (orange) at onset (left) or during (right) a δ1-wave (blue). Unfiltered signals are presented in gray. Values are mean ± s.e.m. indicated as either error bars or shaded areas. In **a**, **b**, and **d** significant changes between 1st–2nd and 2nd–3rd quantiles are indicated with red and black bars, respectively.

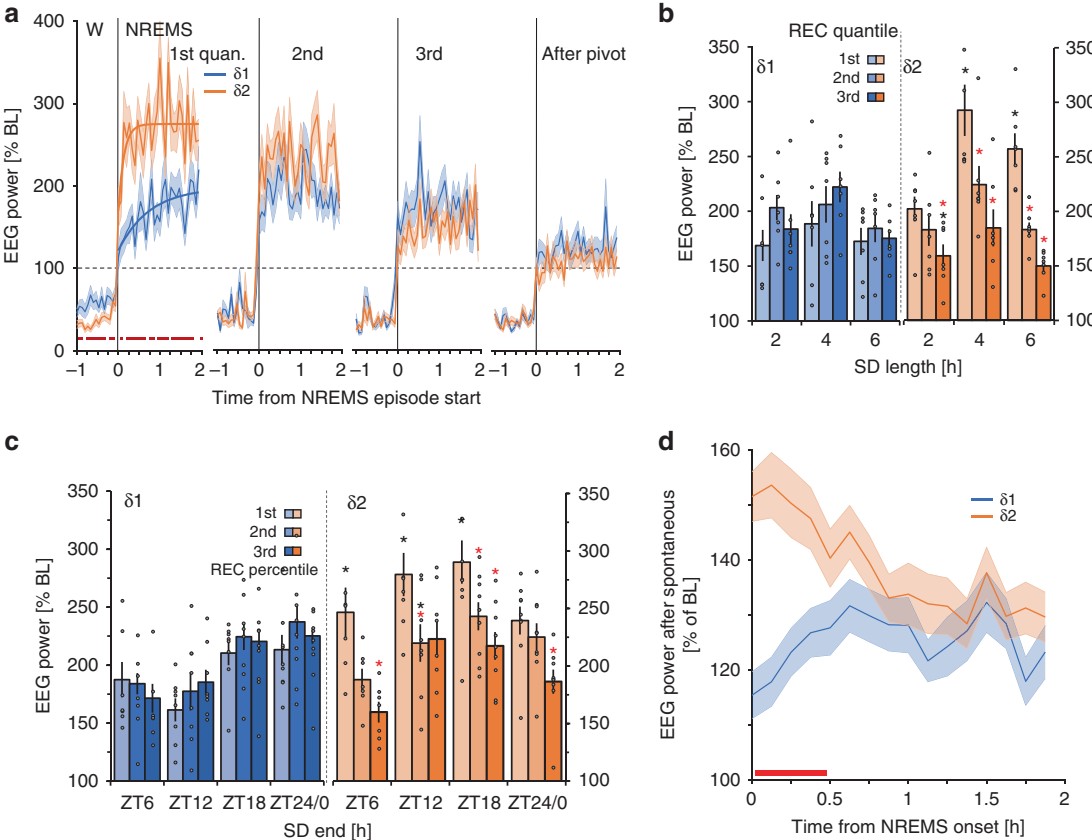

**Fig. 3 δ1- and δ2-dyamics following enforced and spontaneous extended waking. a** δ1- and δ2-dynamics across wake (W) to NREMS transitions during recovery from 6 h SD differed (red squares represent significant differences; paired t-tests, $p < 0.05$, n = 38). Lines represent regressions to saturating exponential functions fitted to 4 s values starting at NREMS-episode onset (time constants: δ1 = 44.8 s, δ2 = 8.1 s). **b** SDs starting at light onset (ZT0) of different lengths (2-, 4-, and 6 h; n = 7/group). δ1 showed an effect only across subsequent quantiles (one-way rANOVA factor time: $F_{2,36} = 9.2$, $p = 0.0006$), whereas δ2 was sensitive to the length of SD (two-way rANOVA factors time × SD length: $F_{4,36} = 5.0$, $p = 0.003$), and different from δ1 (three-way rANOVA factors time × SD length × sub-band: $F_{4,72} = 3.2$, $p = 0.02$). Asterisks represent post-hoc significance ($p < 0.05$) for differences from first quantile (red), or for δ1 vs. δ2 (black). **c** Effect of time-of-day at which SD ends on δ1 and δ2. Asterisks same as in **b** (one-way rANOVA factor SD-end; δ1: $F_{3,30} = 3.86$, $p = 0.02$; δ2: $F_{3,30} = 3.92$, $p = 0.02$; n = 8/group). Black circles represent individual datapoints (**d**) NREMS following prolonged spontaneous waking (1.6 ± 0.1 h) during the baseline dark period showed initial increases for δ2-power which then decreased across subsequent NREMS episodes, whereas δ1 did not (two-way rANOVA factors sub-band × time: $F_{15,1110} = 4.9$, $p = 2.2E{-}09$). Red lines in **d** (post-hoc tests, $p < 0.05$, n = 38). Data is presented as mean values ±s.e.m. indicated as either error bars or shaded areas.

compared to δ1 with differences between the bands dissipating afterwards (Fig. 3a).

Next, we quantified δ1- and δ2-power dynamics in the first three recovery quantiles (24.8 ± 0.9 min NREMS) following SDs

of 2-, 4-, or 6 h. We found that δ1-power immediately after SD was unaffected by differences in preceding SD duration, whereas δ2 showed high values that increased further for the longest two SDs (Fig. 3b). Independent of SD duration, δ2 rapidly declined

during recovery sleep, while a significant increase was observed from the 1st to 2nd recovery quantile for δ1-power. Thus, only δ2-power was in a quantitative relationship with prior time-spent-awake.

To assess whether time-of-day affected recovery dynamics, we reanalyzed a previously published dataset, where 6 h SDs were performed starting at either ZT0, −6, −12, or −18[27]. In this experiment NREMS-onset after SD could not be controlled, and NREMS-onset latency after the SD ending at ZT18 amounted to 2.8 h thereby extending time awake to ~9 h[27]. Initial values of δ2-power reached their highest levels during this experiment as a consequence of this increased prior wake time (Fig. 3c). Power in the δ2-band decreased in the subsequent two quantiles at all times-of-day. In contrast, initial δ1-values did not peak after the SD ending at ZT18, and its levels followed a distinctly different time-course, with higher levels reached after SDs ending at Z18 and ZT24/0 and the lowest after ZT12 (Fig. 3c). In the remaining two recovery quantiles, δ1 did not consistently change but increased from its initial levels at most times-of-day. These results suggest that while δ2 is sensitive to prior time-spent-awake, initial increases in δ1 may be sensitive to time-of-day.

We next considered whether spontaneous prolonged waking bouts similarly provoked differential δ-dynamics. Baseline dark periods began on average with such waking periods (1.6 ± 0.1 h), followed by consolidated NREMS, at a time where we had previously observed SW-period shortening (Fig. 1d). Power in both δ sub-bands were increased above baseline levels (ZT8-12) although significantly higher for δ2 compared to δ1 (Fig. 3d). Surprisingly, although δ1 was above baseline reference values, it did not decay during NREMS whilst δ2-power did. Through these three experiments, we found that both δ1- and δ2-dynamics, as well as the differences between them, were consistent and robust. Although δ1 always increased after SD, it was smaller than δ2 and importantly, not in a quantitative relationship with prior wake duration and did not decrease over the subsequent two quantiles of NREMS, which are considered defining criteria for homeostatically regulated sleep variables.

**Frontal cortex and the thalamic CMT shape δ2-dynamics**. The data presented above were obtained using our standard frontal-parietal derivation[28]. To determine if δ1- and δ2-dynamics differed according to cortical area, we implanted a subset of animals with frontal, central, and parietal cortical electrodes referenced to the cerebellum. Consecutive to long periods of spontaneous or enforced waking, frontal dominance was observed solely for δ2, while δ1-dynamics did not vary across the cortical surface (Fig. 4a). As the greatest δ2−δ1 difference among derivations was identified during the first NREMS quantile of SD recovery (7.7 ± 1.4 min NREMS), we compared EEG spectra at this time to ascertain the frequency specificity of this effect. The largest differences were seen in the δ2-frequency range, especially between frontal and parietal derivations (Fig. 4b). These analyses confirmed that the frontal dominance of δ-power under conditions of elevated sleep pressure was specific to a narrow frequency band encompassing δ2[29].

To gain insight into the possible circuitry underlying δ1- and δ2-generation, we probed deeper cortical and thalamic regions, using a published dataset[30] of local field potential (LFP) recordings at three cortical (visual, barrel, and cingulate) and two thalamic sites (centromedial thalamus (CMT), anterodorsal nucleus (AD)). Following a 4-h SD ending at ZT4, high levels in both δ1- and δ2-power were observed in visually similar patterns to cortical surface electrodes and strikingly similar across recording sites (Fig. 4c). δ2-power rapidly decreased across all sites, while δ1-power initial increased before linearly decreasing.

Finally, we found that during recovery, ON-/OFF-state lengths recorded from the cingulate cortex, were initially shorter and returned to stable levels as sleep progressed (Fig. 4d), reminiscent of the shortening of SW-period after SD (Fig. 1d). Spiking rates across recovery did, however, not change and were consistently greater for ON- than OFF-states (12.8 ± 2.9 and 3.9 ± 1.2 Hz, respectively).

The CMT is thought to be a critical component of the circuit underlying SW-generation during NREMS[31]. Optogenetic silencing of CMT neurons (Fig. 5a) during recovery sleep, was previously shown to increase δ-power in NREMS episodes subsequent to silencing, in the cingulate cortex, which receives direct input from the CMT[30]. We found that this δ-rebound after CMT inhibition was specific to the δ2-band in the cingulate and visual cortices though not in the barrel cortex (Fig. 5b, right). In all three structures, δ1-activity remained unaffected (Fig. 5b, left). During 10 s of optogenetic silencing of CMT neurons in baseline, both δ1- and δ2-activity was suppressed, though only the latter rebounded over pre-silencing levels immediately afterwards (Fig. 5c). Both the suppression and the rebound which followed were specific for the cingulate and visual cortices and not observed in the barrel cortex (Fig. 5c). This lack of effect on either δ1- or δ2-dynamics in the barrel cortex is consistent with our previous results, demonstrating that this structure is out-of-circuit from higher order thalamocortical networks during NREMS[30].

**δ sub-band-specific responses to sleep deprivation in humans**. Using previously published data sets[32–35], we next assessed whether humans showed a similar δ-power heterogeneity in response to SD. A total of 110 healthy human subjects kept awake for 40 h were included in the analysis. Using a similar approach as with mice, we focused on spectral dynamics in the right fronto-parietal (F4-P4) derivation, over consecutive NREMS episodes. δ-dynamics appeared heterogeneous, with lower frequencies initially increasing, and faster ones decreasing (Fig. 6a). Using these results, we obtained as representative bands 0.5–1.0 Hz and 1.5–2.0 Hz, for δ1 and δ2, respectively. After SD, δ2-power was initially high and decreased rapidly across the first two NREMS episodes, while δ1 reached lower levels than δ2, which subsequently increased in the second NREMS episode vs. the first (Fig. 6b). Similarly, during the first NREMS episode of baseline sleep, δ1 was initially lower than δ2, while the opposite was true during the second episode (Fig. 6b). Thus, as in mice, δ2-power in humans reach higher maximum values than δ1 during the first NREMS episodes for baseline and recovery sleep followed by steeper decays than δ1.

To determine whether δ-dynamics were region-specific or represented a global cortical process, a subset of individuals (n = 21) with 19 recording sites were analyzed[35] (Fig. 6c). After SD, the highest relative δ2-power increases were confined to frontal areas, whereas δ1 was more globally diffuse (Fig. 6d), similar to the δ2-specifc frontal dominance observed in mice. In the first NREMS episode, vast differences were observed between the two δ-bands, which dissipated across subsequent episodes. During the second NREMS episode some significance persisted in frontal areas, though δ2 decreased faster in more posterior areas falling below relative δ1-levels (red circles). Our results indicate that differential δ1- and δ2-dynamics are similarly present in humans, although both δ-bands center at lower frequencies than in mice. In both species, δ2-activity was more sensitive to sleep-wake history with frontal cortical areas showing largest dynamics.

**δ2-recovery dynamics parallel physiological changes**. The short-lived nature of the δ2-rebound after SD was evidenced in mice

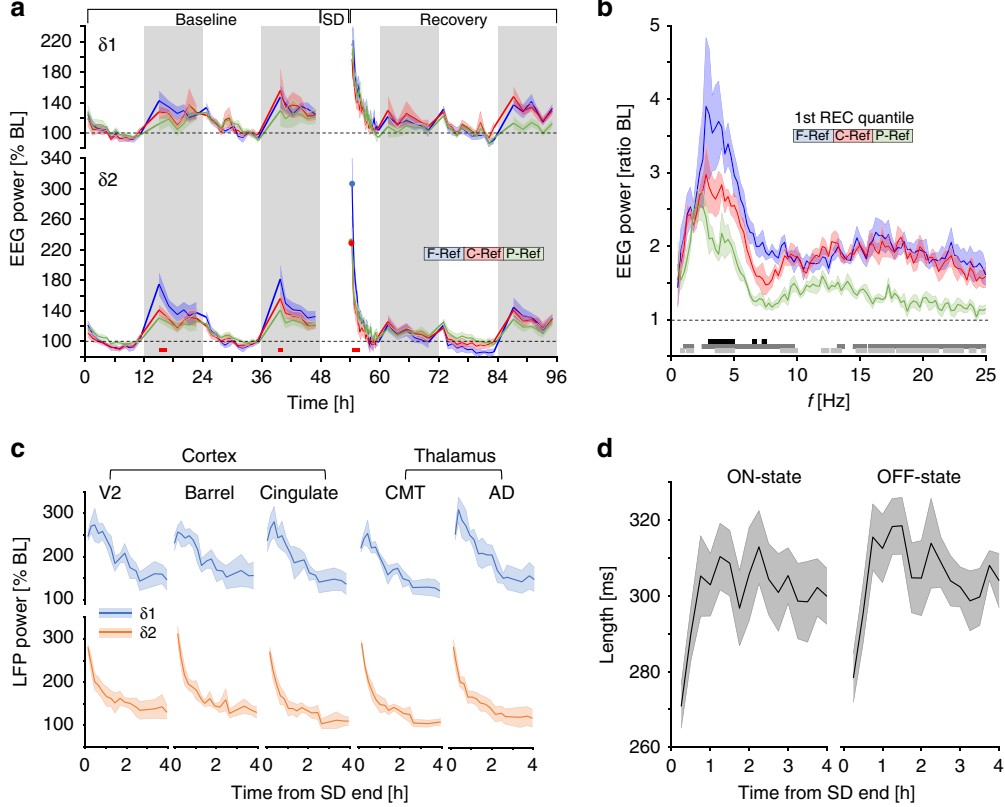

**Fig. 4 δ2-dynamics are visible throughout the thalamus and cortex. a** Time-course of δ1- (upper) and δ2-power (lower graph) at different cortical surface sites. δ2 is high at specific times during baseline (dark period) and following SD, and was highest in frontal areas, whilst derivation did not affect δ1 (δ2: 2-way rANOVA factors site × frequency bin: $F_{168,1008} = 1.7$, $p = 1.0E-6$). Circles denote initial values following SD for δ2 to emphasize frontal dominance in rebound. Red squares denote post-hoc significance between F-Ref and C-Ref/P-Ref. **b** EEG spectra differed across recording site during initial recovery NREMS (two-way rANOVA factors site × frequency bin: $F_{196,1470} = 1.44$, $p = 2.0E-4$). Black, dark gray, and light gray lines in lower panel represent post-hoc significant differences ($p < 0.05$; Tukey) between F–C, F–P, and C–P, respectively ($n = 6$). **c** LFP recordings following 4 h SD in multiple brain structures showed similar dynamic changes in both δ1- and δ2-power as in **a**. **d** Average length of ON- (left) and OFF- (right) states during the 1st 4 h of SD recovery as in **c** in the cingulate cortex shows immediate decreases after SD end (one-way rANOVA factor time; ON-state: $F_{15,45} = 5.39$, $p = 0.00001$; OFF-state: $F_{15,45} = 7.06$, $p = 1.5E-07$). Values are presented as means ± s.e.m.

where nearly half of the initial power was lost during the first hour of recovery (δ2$_{initial}$ = 261 ± 13%, δ2$_{1h}$ = 136 ± 3% of reference) and between NREMS episodes 1 and 2 in humans (δ2$_{1st\_episode}$ = 369 ± 12%, δ2$_{2nd\_episode}$ = 233 ± 9%; Figs. 2f, 6b; Supplementary Fig. 6b). Given this observation, is sleep homeostasis as evidenced by δ-dynamics truly a continuous process, or does the non-continuous rapid decay in its faster and main component point to a discrete sub-state of NREMS, emerging after prolonged periods of wakefulness? During this early recovery phase in the mouse, a number of additional variables initially attained values atypical of NREMS that all reverted to baseline levels before the δ2 pivot-point. We already noted that spectral power of higher EEG frequencies during NREMS (beta/low-gamma 18–45 Hz) was significantly increased during initial episodes of recovery (Fig. 2d) and when these frequencies were analyzed identically to δ2-power, a familiar pattern emerged (Fig. 7b). Even more surprisingly, other physiological measures followed similar dynamics. Muscle tone during NREMS was initially high, followed by a rapid decrease before reaching stable baseline values after pivot-point (Fig. 7c). Moreover, cortical temperature ($T_{cortex}$), recorded in a subset of mice, showed a significant reduction up to pivot-point, losing >2 °C in <60 min (Fig. 7d). However, changes in respiratory rate during NREMS did not covary with these other measures (Fig. 7e).

Some of these variables were similarly affected after periods of spontaneous waking in baseline. NREMS EMG was also increased immediately after sleep onset during the dark periods (Supplementary Fig. 6a), in conjunction with δ2-dynamics but not δ1 (Supplementary Fig. 6a, b). This wake-history-dependent difference was further corroborated by analyzing δ2/δ1-power ratio, which saw the lowest values during the majority of the light periods, followed by increases in baseline dark and after SD (Supplementary Fig. 6c). Examination of EEG dynamics in other frequency bands (theta, sigma, and beta/low gamma) during NREMS, showed similar behavior (Supplementary Fig. 6d).

Changes in brain temperature affect the frequency of EEG oscillations such as theta (6–9 Hz) and the SO[36,37]. To determine if temperature was causally related to δ2-dynamics, we maintained $T_{cortex}$ at SD levels during the first hour of recovery using a thermoelectric generator without affecting sleep-wake state (Fig. 7f, Supplementary Fig. 7). However, impeding the $T_{cortex}$ decrease did not affect the recovery time-course of δ2 (or δ1)-power (Fig. 7g). Another example illustrating this lack of causality among the highly correlated physiological variables, is that increasing δ2 through CMT silencing did not significantly alter EMG dynamics during recovery sleep (SD controls: 146 ± 13%, SD with CMT silencing: 167 ± 18%, two-way rANOVA factors condition × time-course: $F_{7,42} = 0.25$, $p = 0.92$).

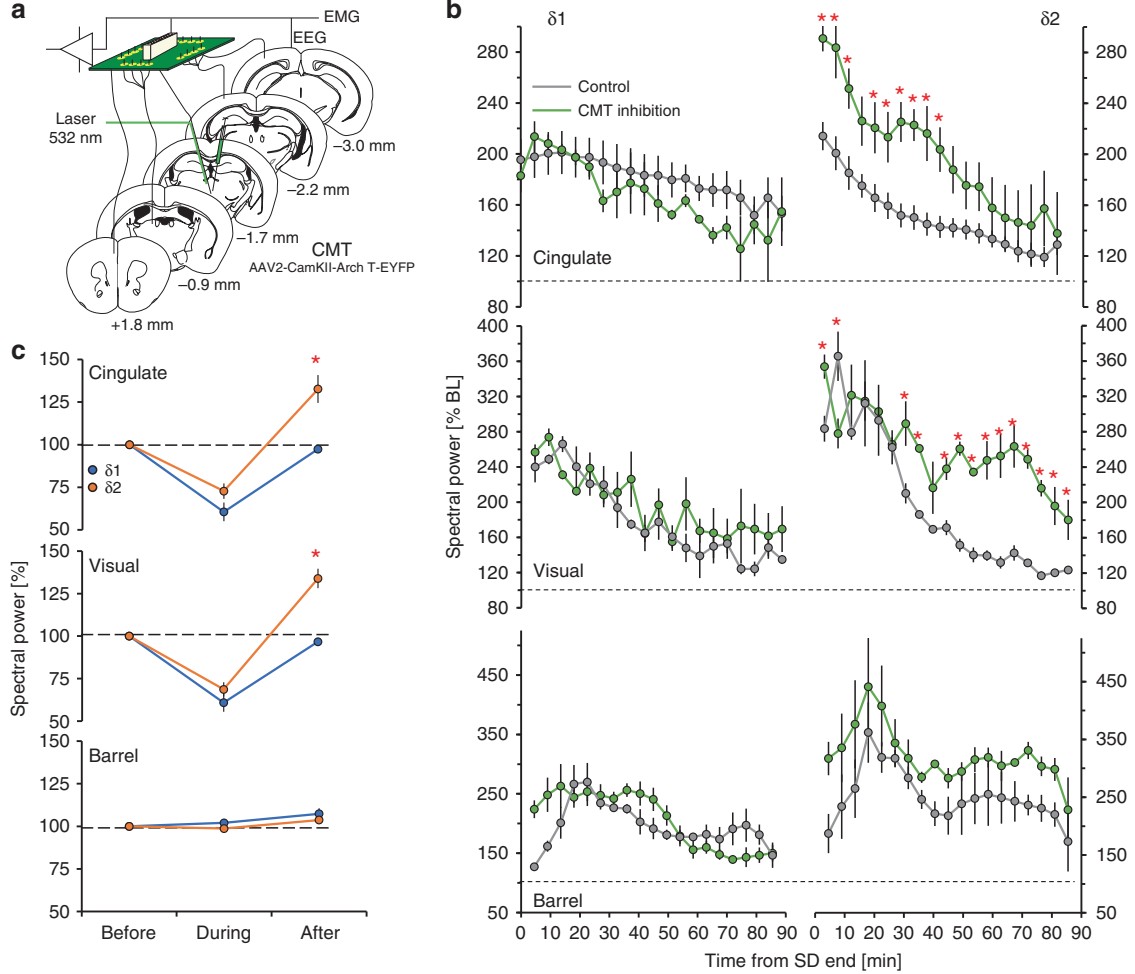

**Fig. 5 Thalamic silencing alters δ2-dynamics in cingulate and visual cortex. a** Schema showing optogenetic inhibition of the CMT. Certain elements were re-used from a previous publication by the same authors[30]. **b** increased δ2-power (right) during NREMS (green) following a 4-h SD in the cingulate (top) and visual (middle) cortices, but not the barrel cortex (bottom) as compared to control SD animals (gray) (two-way rANOVA factors silencing × time; cingulate: $F_{18,108} = 2.4$, $p = 0.003$; visual: $F_{18,108} = 4.7$, $p = 1.8E-07$). δ1-power (left) was unaffected in all structures by CMT silencing. **c** Mean δ1- and δ2-power in these structures in baseline before, during, and after 10 s optogenetic silencing during NREMS of ArchT-expressing CMT neurons (one-way rANOVA factor 'silencing'; cingulate: $F_{1,8} = 73.7$, $p = 3.0E-4$; visual: $F_{1,8} = 48.4$, $p = 1.0E-4$). Red asterisks denote significant post-hoc differences from controls ($p < 0.05$). Values are expressed as mean ± s.e.m.; in **b** $n = 4$, in **c** $n = 6$.

In humans, we observed pronounced changes in the alpha-frequency range (8–11 Hz) in the first, relative to the second, recovery NREMS episode (Fig. 6a). Time-course analysis revealed that SD greatly increased alpha activity specifically during the 1st NREMS episode (Supplementary Fig. 6e), as previously noted[38]. Moreover, in a subset of subjects recorded with ECG ($n = 44$), we observed that heart rate levels and dynamics during the first recovery NREMS episode differed markedly from episodes that followed and from baseline (Fig. 7h), confirming previous findings[39]. The uniqueness of the first NREMS episode after SD was further reflected in heart rate variability (Fig. 7h, insert), considered a measurement of sympathovagal tone[40]. These observations are in line with previous studies noting similarly high levels in other physiological measures during the first NREMS episode, such as EMG[24] and body temperature[41]. These sets of physiological variables with comparable fast dynamics in the first hour for recovery in the mouse or the first NREMS episode in humans, demonstrates that early NREMS qualitatively differs from all subsequent NREMS. This NREMS phase could be described as a phase of wake-inertia during which a wake-like physiology coincides with deepest NREMS.

## Discussion

Through in-depth analyses of SW-features in mice, we noted that during NREMS immediately following extended periods of waking, either spontaneously- or experimentally-evoked, δ-waves briefly increased in frequency. We discovered that this was not due to a general shortening of SW-periods across the δ-band, but instead to an increased incidence and amplitude of SWs belonging to a population of faster δ-waves (2.5–4.5 Hz), which we termed δ2. Prevalence and amplitude of a slower population (0.75–2.0 Hz; δ1) remained relatively unperturbed during this initial NREMS phase and preceding time-spent-awake did not predict their initial levels. Perhaps the most salient result of our study is the short-lived nature of δ2-wave promotion, as within the first hour following SD, their features (power, amplitude, and incidence) reverted to near baseline levels. The δ2-recovery time-course was paralleled by an equally fast decay in the levels of a number of physiological variables not considered to be associated with the process of sleep homeostasis. We interpret this highly dynamic NREMS phase to reflect a phase of wake-inertia during which the system adjusts from the aftereffects of a highly active and sustained waking period before typical NREMS can be reinstated.

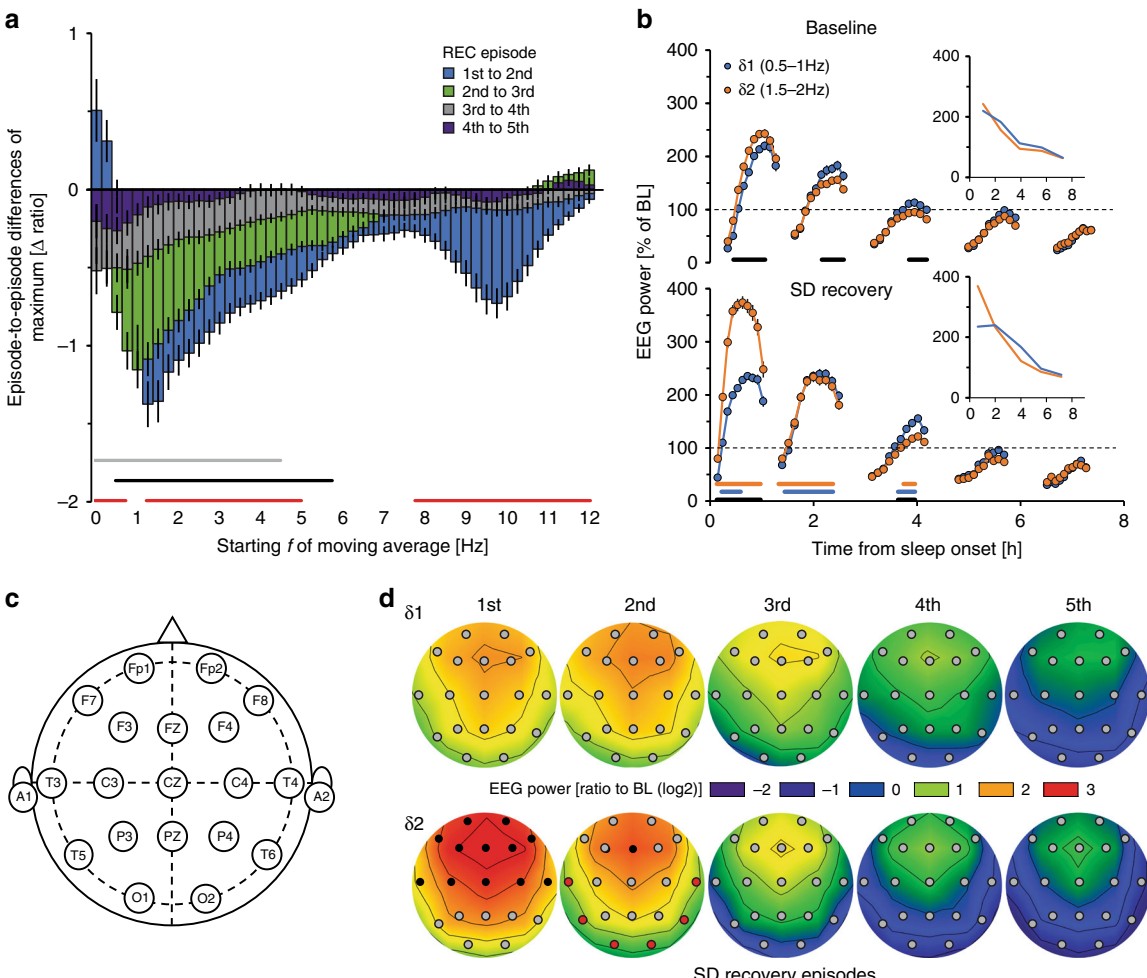

**Fig. 6 δ-band heterogeneity in humans. a** Differences between consecutive recovery (REC) NREMS episodes in highest EEG power reached per episode [three 0.25Hz-bin moving average per quantile; see B; 1st–2nd (blue), 2nd–3rd (green), 3rd–4th (gray), and 4th–5th (dark blue)] following a 40-h sleep deprivation (SD). The largest decrease was observed in faster δ (1.5–2.0 Hz), while activity in slower δ (0.5–1 Hz) increased initially (two-way rANOVA factors episode-to-episode differences × frequency: $F_{48,10464} = 13.6$, $p = 1.0E-17$, $n = 110$). Red, black, and gray lines represent post-hoc significance ($p < 0.05$), 2nd/1st, 3rd/2nd, and 4th/3rd NREMS episode, respectively. Data is expressed as means ± s.e.m. **b** Time-course plots of δ1- (0.5–1.0 Hz) and δ2- (1.5–2.0 Hz) power during each NREMS episode (10 quantiles/episode) in baseline conditions (BL; top) and recovery (bottom). Gaps reflect intervening wake or REMS. During BL, significant δ1-δ2 differences were seen (two-way rANOVA factors sub-band × time-course: $F_{49,10437} = 8.6$, $p = 1.0E-17$). After SD, δ2 was increased during the first NREMS episode, illustrated by the time-course of maximum values per episode (insert; REC: two-way rANOVA factors sub-band × time-course: $F_{49,10437} = 33.6$, $p = 1.0E-17$). δ1-δ2 differences were seen during both BL and recovery periods (two-way rANOVA factors SD × time-course; BL: $F_{49,10486} = 4.1$, $p = 1.0E-17$; REC: $F_{49,10388} = 29.2$, $p = 1.0E-17$). Black lines represent post-hoc significance (Tukey) between quantiles of δ1 and δ2, blue and orange between δ1 and δ2 from baseline to recovery, respectively. Values are given as percentage of baseline mean ± s.e.m. (black lines; $n = 110$). Data are based on the right frontal-parietal derivation (F4-P4). **c, d** Electrode placement, and heat plots in a subset of subjects ($n = 21$) with 19 electrode sites, of δ1- and δ2-power during REC. Increases in δ2 were highest in frontal areas and dissipated more quickly in subsequent NREMS episodes than δ1 (two-way rANOVA factors electrode site × sub-band; 1st NREMS episode: $F_{18,576} = 5.7$, $p = 1.6E-12$; 2nd: $F_{18,648} = 4.3$, $p = 8.2E-09$). Values are expressed as log2 of ratio to mean BL power across all electrode sites, for each subject. Black and red filled circles represent electrode sites where δ2 was significantly higher and lower, respectively, than δ1 (Tukey post-hoc $p < 0.05$). Black contour lines are fixed at 0.5.

Many have noticed heterogeneity in the sleep-wake dependent changes in the δ-frequency range[1,7,29,42–44]. In one of the first all-night spectral analyses of the human sleep EEG, one can already observe that at 1 Hz EEG power does not decrease from the first to second NREMS episode[14]. Spectral analyses can, however, not assess the various SW-attributes nor can it give insight into the temporal organization of different wave elements in the signal. The bimodal distribution of SW-incidence across frequencies that could be observed throughout the recording period, the differences in the waveforms of δ1- and δ2-waves, and their different dynamics in response to sleep-wake history, together argue for the existence of two classes of SWs.

Results of three unbiased methods converged on ~2.0/2.25 Hz separating the sleep-wake driven dynamics in the δ1- and δ2-bands in the mouse, which matched remarkably well the results of other rodent studies[7,29,42,45]. With a similar approach we determined ~1.0/1.25 Hz as the frequency separating the two δ-bands in humans; i.e., left-shifted by 1 Hz compared to the mouse. What we refer to as δ1 in humans falls in the frequency range generally reserved for the SO[46], the dynamics of which, similar to our findings, did not decrease during the first NREMS episode[47]. Applied to the mouse, SO's upper demarcation might be as high as 2.5 Hz, as others have already assumed due to other considerations[48,49]. Accordingly, the δ2-population of SWs could

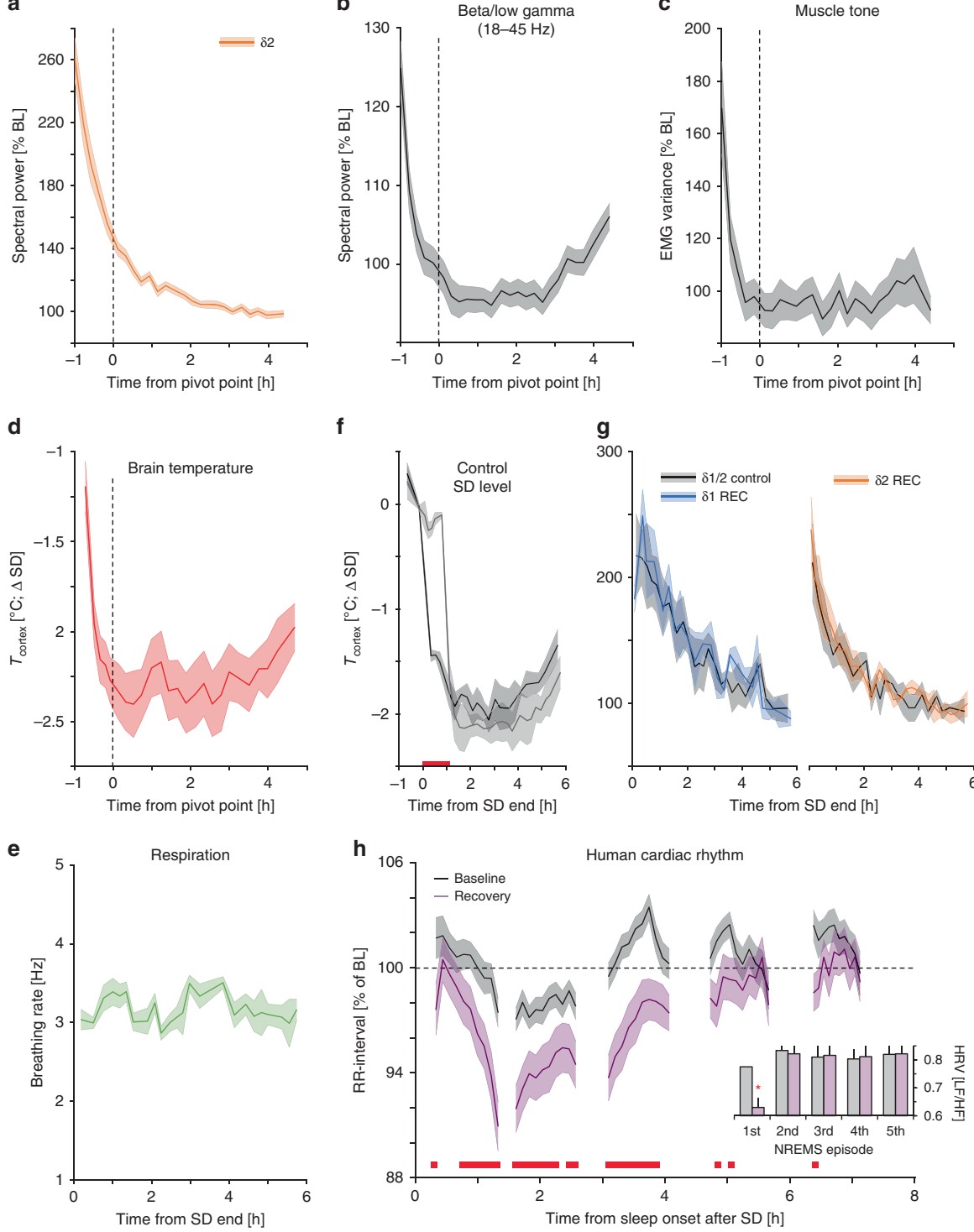

**Fig. 7 δ2-dynamics typify a physiologically different NREMS sub-state. a** δ2-power declines across the 1st 6 h of recovery as does **b** EEG power in higher frequencies (beta/low-gamma: 18–45 Hz), **c** muscle tone, and **d** brain temperature ($T_{cortex}$), but not (**e**) respiratory rate. Of note, beta/low-gamma power was greater following SD than at any point during baseline (Supplementary Fig. 6f). **f, g** Experimentally maintaining $T_{cortex}$ at sleep deprivation (SD) levels during the first hour of recovery (REC) in a subset of mice ($n = 4$), did not alter δ1- or δ2-dynamics. **h** Human cardiac rhythm (measured as RR-intervals expressed as percentage of mean baseline (BL) RR-interval (1.07 ± 0.02 s)) differed early in NREMS during recovery compared to baseline, as well as subsequent episodes (two-way rANOVA factors condition × time-course: $F_{49,3822} = 1.8$, $p = 0.001$), which was also reflected in heart rate variability (insert: HRV quantified as the LF/HF-ratio (see Methods); two-way rANOVA factors condition × time-course: $F_{4,312} = 6.7$, $p = 0.008$]. All values are mean ± s.e.m. and post-hoc significance ($p < 0.05$) is denoted with red lines (or asterisk; H-insert). Vertical dashed lines in **a–e** represent average of individually calculated δ2 pivot-points. **a–c** ($n = 38$), **d** ($n = 8$), **e** ($n = 3$), **f, g** ($n = 4$), **h** ($n = 44$).

represent δ, albeit with species-specific lower- and upper-frequency boundaries. Species differences have been largely overlooked as similar frequencies are used to delimit the δ-band across mammals and birds. The SO, characteristic of cortical LFP or EEG signals during NREMS, reflects the rhythmic alternation of active (ON or UP) states, during which neurons are depolarized and show maximal firing, and silent (OFF or DOWN) states, periods of hyperpolarization and relative quiescence. The SO UP-state is thought to group the occurrence of δ-waves and spindles in the cortico-thalamic network[21,50,51]. Our observation of δ2-waves nesting in almost all δ1-waves, thus adds further credibility to δ1 representing the SO.

Narrowing the frequency range best gauging prior wakefulness, as well as the time required to relax back to baseline during NREMS puts constraints on the possible neuronal substrates underlying the sleep homeostatic process SW-generation is thought to reflect. The sleep-wake driven dynamics based on which we identified the two SW-types have rarely been taken into account when studying the cellular and network substrates of SWs generated in the thalamocortical circuitry. Directly relating our findings to in vitro work or studies using anaesthetized, awake, or immobilized animals recorded out of a dynamic sleep-wake history context might therefore not always be straightforward. We found that the well-known frontal dominance of δ-power after SD[52], was specific to the δ2-band in both species, whereas δ1 increases were smaller and concerned a larger surface area of the cortex, a finding also observed recently in humans[53]. This latter observation is consistent with the SO engaging the entire neocortex[21,54,55]. Indeed, the SO was first described as a cortical phenomenon with layer-5 pyramidal neurons being key to UP-state initiation[21,54,56], while for δ-waves two sources have been described, one a clock-like oscillation generated by the intrinsic properties of thalamocortical neurons when hyperpolarized[51,57], and another of cortical origin likely to be generated in a similar fashion to cortical UP- and DOWN-states characteristic of the SO[21]. Thalamocortical crosstalk is an important contributor to the generation of both the SO and δ-waves[21,58]. For instance, cortical UP-states synchronize thalamic δ-oscillations[51], which might contribute to the prominent nesting we observed of δ2-waves at δ1-wave onset, while excitatory thalamocortical input to the cortex can trigger cortical UP-state initiation and removing thalamic input reduces cortical SO-period and -synchrony[30,59,60]. Such observations demonstrate that SWs must be regarded as an emerging property of the thalamocortical network acting as a single unit[58]. Consistent with this, our in vivo LFP recordings showed that the dynamics typical of each of the two SW-populations could be observed throughout much of the cortex and thalamus with little variation precluding identification of their primary respective source.

In an effort to determine a thalamic contribution to either SW-population, we optogenetically inhibited the CMT, a non-sensory nucleus important in the timing and manifestation of cortical SWs[30]. CMT inhibition boosted δ2-activity specifically but whether this augmentation is of thalamic or cortical origin is unclear as we quantified SWs during NREMS episodes consecutive to those during which the CMT was silenced, indicating that increases in δ2-activity are a cortical rebound phenomenon[30]. Alternatively, CMT silencing might have affected local circuitry dynamics recruiting other thalamic nuclei into modulating δ2-activity, or the suppression of SWs during CMT silencing could have released δ2-waves from the constraints imposed on their occurrence by δ1-waves (i.e., nesting). Interestingly, this δ2-increase and lack of δ1-change was also observed in the visual, though not the barrel cortex, confirming that the barrel cortex is external to an N-type thalamo-cortico-thalamo-cortical circuit consisting of the CMT-cingulate-AD-visual-cortex excitatory pathway[30].

Optogenetic stimulation of inhibitory thalamic reticular neurons and pharmacological inhibition of the excitatory somatosensory thalamus, were both shown to induce cortical SWs at δ2-frequencies, confirming our CMT results, although it should be noted that these results were obtained in awake mice[61,62]. Thus, our current data do not allow for the dissecting of the precise elements in the thalamocortical circuitry underlying δ1- and δ2-generation. Outside of the thalamic involvement we established, a cortical contribution, especially for δ1-activity, is more than likely. Our data also underline the need for studying SW-generation in a dynamic and in vivo context, to better elucidate the cellular and network substrates of the sleep-wake driven changes in SWs.

The high δ2-levels reached during NREMS immediately after SD were accompanied by high levels of a number of physiological variables more reminiscent of waking than of NREMS, suggesting that during this time it carries signatures of preceding wakefulness setting it apart from subsequent sleep episodes. Paradoxically, these early NREMS episodes are considered the deepest and most recuperative[63,64].

The process of transitioning from wake to sleep is characterized by a short-lived (several minutes) coexistence of wake- and sleep-like EEG patterns, indicating that the timing of falling asleep varies among brain structures, with the thalamus becoming deactivated prior to cortex, and fronto-central cortical areas prior to more posterior regions[43,65–67]. Our data extend these observations to physiological variables such as muscle tone, heart rate, and temperature during recovery and also demonstrate that under these conditions this transition phase is prolonged, encompassing the entire first NREMS episode in humans and the time up to the pivot-point in mice. This is strikingly illustrated in the mouse where NREMS with the highest SW-activity (i.e., deepest sleep) coexists with exceptionally high brain temperature, muscle tone, and beta/low-gamma EEG levels, all atypical for this state.

The co-existence of incongruous sleep- and wake-like features is, in some respects, reminiscent of NREMS parasomnias, including sleep-walking, that most often occur during the first NREMS period when sleep is deepest and, at the level of the EEG, is associated with increases in δ-activity in frontal areas and increases in wake-like EEG activity in the alpha- and beta-bands in sensorimotor areas[68–70], perhaps related to the high alpha-activity we observed in the first recovery NREMS episode. Thus, the high initial δ2-levels and the subsequent steep decline might not only be emblematic of this transient NREMS phase, but also instrumental in facilitating the passage from a highly active brain state and physiology to typical NREMS. Through deactivating the thalamus and frontal cortex, this would prevent conscious experience of a still awake system, as is the case during sleep-walking. The finding that EEG δ-activity during anesthetic-induced loss-of-consciousness shares features with those of early NREMS after SD, is consistent with this line of thought[71,72].

The dynamics of the sleep homeostatic process, aka Process S[10,13], were derived from the sleep-wake dependent changes in power in the full δ-band. These dynamics show gradual and continuous changes in relation to sleep-wake distribution and can be reliably modeled by assuming exponential saturation functions for the increase during wake and decrease during NREMS. Here we show that this gradual time-course, most strongly exemplified in the mouse, is an artefact of combining the activity of two SW-populations with very different sleep-wake driven dynamics. Because δ1-activity was not in a quantitative relationship with SD duration, did not increase after spontaneous wakefulness, and did not decrease (in humans) or even increased (in mice) during initial recovery NREMS, it cannot be considered a proxy of an underlying sleep homeostatic process. Nevertheless, SD did

increase δ1-activity compared to baseline in the mouse. This δ1-increase could relate to aspects of extended wakefulness that do not monotonically increase with time-spent-awake. For example, it was found that the SD-associated increase in δ1-activity depends on cortical noradrenalin levels and that these levels are elevated with SD but, especially in frontal cortical areas, already peak after 3 h before again decreasing towards the end of SD[73,74]. Conversely, δ2-power and δ2-wave incidence and amplitude, all increased as a function of time-spent-awake and decreased during recovery sleep, suggesting a homeostatic mechanism. δ2-activity measures were, however, depleted within 1 h, belying its functional necessity to continue sleeping.

Sleep-wake dependent changes in specific SW-properties, are integral to current leading hypothesis on sleep function stipulating that wakefulness is accompanied by synaptic strengthening that needs to be homeostatically balanced during NREMS through a process referred to as synaptic downscaling[12]. In this context, steeper SW-slopes, independent from changes in SW-amplitude, are thought to reflect more synchronized and rapid transitions between neuronal ON- and OFF-states due to increased synaptic strength[25,26]. We found that increased amplitude and incidence of δ2-waves fully account for the changes in SW-slope observed after SD with close to 1.0 correlations between slope and amplitude for SWs > 2.0 Hz (Fig. 2c) even after correcting for SW-amplitude (Supplementary Fig. 8). Similar findings were observed after SD in humans albeit for SWs > 1 Hz[75], consistent with the 1 Hz lower δ1-δ2 demarcation in humans versus mice. If δ2-activity were to reflect a synaptic downscaling process, then it is accomplished within 1 h. Although we cannot establish whether the process requires this little time and might indeed be an important function of this initial phase of sleep, the similarity of the discontinuous time-course of δ2 with that of physiological variables not associated with the sleep homeostatic process makes it more plausible that these changes are part of transition phase before entering a more consistent resting state. These short-lasting and discontinuous dynamics (exemplified with the pivot-point), are inconsistent with accepted models of sleep regulation and function based on those of a merged δ-band as a sleep-need proxy.

To conclude, the discovery of two SW-populations and their respective sleep-wake dependent dynamics concomitant to changes in a number of physiological variables, implies the existence of a transient NREMS state representing a mere ~3% of total NREMS in the mouse across a 96-h experiment. We believe that this early and fleeting portion of NREMS, which is enhanced by SD, serves as a nexus between an active waking state and subsequent quiescent NREMS. The large, wake-dependent increases in δ2-activity during this transition phase could still serve a homeostatic function insomuch as it gates the transition into sleep. Such a scenario would allow us to refocus attention on the regulation and function of time-spent-in-NREMS and REM sleep, as their rebounds after SD have not received much attention in current hypotheses on sleep regulation and function.

## Methods

**Mouse experiments**. All mice were C57BL/6J males, aged 10–16 weeks at time of surgery. After surgery mice were housed individually, at a constant ambient temperature (25 °C) and under a 12-h light/12-h dark cycle (LD12:12, fluorescent lights, intensity 6.6 cds m⁻², with zeitgeber time (ZT)0 and −12 designating light- and dark-onset, respectively) and with food and water given ad libitum. All animal procedures outlined were carried out using the guidelines of Swiss federal law and were preapproved by the Cantons of Vaud and Bern Veterinary Offices.

**EEG/EMG/thermistor surgery**. Surgeries were performed according to previously published protocols under deep anesthesia using a Ketamine/Xylazine solution [8% xylazine (Rompun, 2%), 10% ketamine (Ketazol-100, Graeub), and 82% saline (0.9% NaCl)] injected peritoneally (1 ml kg⁻¹) prior to surgical procedures. Body

temperature was constantly monitored using a rectal thermometer and mice were placed on an adjustable heating pad (DC Temperature Control System, FHC). Briefly, mice were implanted with gold-plated stainless steel screws which were soldered to copper wires attached to a connector. Electrodes were placed along an isolateral (right hemisphere) axis on the surface of the cortex (Bregma < 0.2 mm) in the frontal (AP = +1.5 mm, ML = +1.5 mm), central (AP = −1.0 mm, ML = +2.5 mm), and parietal (occipital; AP = −3.0 mm, ML = +1.5 mm), in six mice (Fig. 4a, b). All other mice were implanted with fronto-parietal bipolar electrodes, unless otherwise stated. All signals were differentially referenced with an additional electrode implanted over the cerebellum. Subsequent experiments and datasets used only fronto-parietal electrodes (n = 72). EMG wires were implanted bilaterally in the cervicoauricularis muscles of the neck differentially recorded with one another. Prior to the experiment, animals were given 72 h to recover from implantation, with a further 10 days to habituate to the recording cable. EEG/EMG signals were acquired using a commercially available system (Embla; Medcare Flaga, Thornton, CO, USA). Briefly, signals were amplified, filtered, and analog-to-digital converted to 2000 Hz and downsampled to 200 Hz for analysis. A separate group of mice (LFP, optogenetics experiments) were recorded using the Intan RHD2000 signal processor which sampled all channels at 20000 Hz. Signals were then downsampled to 200 Hz and analyzed in Matlab using custom scripts (see below), and sleep-wake states were annotated with Somnologica (Medcare Flaga, Thornton, CO, USA) using identical criteria. In a subset of mice (n = 9) a thermistor was inserted between the frontal and parietal electrodes to record cortical surface temperature. A thermistor (series P20AAA102M, Thermometrics, Northridge, CA) was inserted through the skull on the surface of the right cortex (2.5 mm lateral to the midline, 2.5 mm posterior to bregma). A constant measuring current 100 μA was supplied to the thermistor and changes in voltage were recorded by the EEG/EMG-acquisition system. Voltage was converted into °C based on the manufacturer-supplied resistance-to-temperature relationship of the individual thermistors.

**EEG/EMG-based discrimination of sleep-wake states**. Following acquisition, signals were filtered for power-line artifacts resulting from AC power cycling (50 Hz). EEG/EMG-based assessment of sleep-wake states was achieved using previously published criteria[28]. Consecutive 4s-epochs were classified as either waking, non-rapid eye movement (NREM) sleep, or REM sleep, using visual inspection without knowledge of the recording condition.

**Thermoelectric generator installation**. To control levels of cortical temperature during SD recovery, in a subset of mice (n = 4) an aluminum heatsink was placed on the skull contralateral to the thermistor (see above, and Fig. 5g), using thermal paste (WLP-1, S+S Regeltechnik GmbH) to ensure efficient temperature transfer. A Thermoelectric generator (TEG; Peltier module: Laird Thermal Systems, Inc. 45850-503¨, Morrisville, USA) was then affixed using dental cement (Paladur) 2 cm above the animal and affixed to the EEG/EMG-recording cable. Temperature manipulation was achieved using a bench top power supply (RND 320-KD3005P, Distrelec, Nänikon, Switzerland) to control amperage input to the TEG. Output temperatures were calibrated prior to the experimentation. Thermal images were acquired using an infrared thermal camera (Model E30, FLIR Systems, Wilsonville, USA) and analyzed using FLIR tools software (v. 6.4, FLIR Systems, Wilsonville, USA).

**LFP tetrode implantation**. Tetrodes were constructed in-lab using four strands of 10 μm twisted tungsten wire, which were then attached by gold pins to an electrode interface board. They were then placed in 2 thalamic and 3 cortical structures: central midline thalamus (CMT) (AP −1.7 mm, ML +1.0 mm, DV −3.8 mm, 15°), cingulate cortex (AP +1.8 mm, ML +0.2 mm, DV −1.6 mm), anterodorsal thalamic nuclei (AD) (AP −0.9 mm, ML ±0.8 mm, DV −3.2 mm), barrel cortex (AP −2.0 mm, ML +2.2 mm, DV −1.1 mm), and visual cortex (AP −3.3 mm, ML +2.5 mm, DV −0.9 mm) and secured to the skull with dental acrylic (C&B Metabond). Optic fibers of 200 μm diameter were placed in the CMT (AP −1.7 mm, ML +1.0 mm, DV −3.8 mm, 15°) and secured with the same dental acrylic. Finally, the implant was stabilized using a methyl methacrylate cement and the animal allowed to recover in the home cage on top of a heating mat. Animals were allowed a minimum of 5 days to recover before starting recordings.

**Optogenetics experiments**. C57Bl6 mice, aged 6 weeks were anaesthetized using isoflurane (1.0–1.5% with oxygen) and placed in a stereotaxic apparatus (Model 940, David Kopf Instruments). Injections of AAV were performed using a 10 μL Hamilton syringe attached to an infusion pump (Model 1200, Harvard Apparatus), in the CMT (AP −1.7 mm, ML + 1.0 mm, DV −3.8 mm, 15°, 100 nL and performed at 0.1 μL min⁻¹ and the needle left in situ for 10 min afterwards to facilitate diffusion. Animals were injected with AAV2-CaMKII-E1fa-ArchT3.0-EYFP (ArchT) for optical silencing of CMT neurons. All plasmids were obtained from University of North Carolina Vector Core Facility. Animals were given 21 days to recover before optogenetic experiments commenced. Animals were instrumented for tetrode recording (see above: LFP tetrode implantation) 3–4 weeks after virus injection to allow sufficient time for opsin expression.

Optical fibers connected to a black furcation tubing coated patch chord (Doric Lenses) were further covered in black varnish to reduce optical leakage from the laser. Animals were habituated to being tethered for up to 8 h per day until a

normal sleep–wake episode resumed based on EEG recordings from ZT4-9. Experiments were performed from ZT4-8, following SD (Fig. 5b). Inhibition was achieved with a green laser (532 nm; LRS-0532-GFM-00100-03, Laserglow Technologies) starting 10 s after NREMS onset. Output was controlled using a pulse generator TTL (Master-9, AMPI or PulsePal 2, Sanworks), co-acquired with all recordings. During sleep deprivation recovery periods, a 10-min moving window was used to calculate δ1 and δ2 power and referenced to the same frequency bins during the preceding baseline day. In a separate experiment (Fig. 5c), optogenetic silencing was performed for 10 s during NREMS between ZT4-8 under baseline (non-sleep-deprived) conditions. δ1 and δ2 power was calculated in 10 s windows pre-, during, and post-inhibition.

**Sleep-deprivation protocol.** After >10 days of recovery from surgery mice were recorded for EEG/EMG for a total of 96 h. For the first 48 h animals remained in undisturbed baseline conditions. Beginning on the third day, mice underwent a 6-h SD starting at light onset (ZT0) using gentle handling[28]. Animals were then allowed to recover for a further 42 h. Other experiments presented in this article also involved SD starting at different ZT across the 24 h period (Fig. 3c[27]), or of varying lengths (2- and 4 h, Fig. 3b).

For LFP recordings and optogenetic experiments, animals were taken from their home cages and placed in new ones at ZT0 with clean bedding, food, and water, in addition to a novel plastic object. A 4-h SD was performed before mice were transferred to their original cages for data acquisition between ZT4 and −9. For more details see[30].

**Respiratory recordings.** To determine breathing rates, mice were recorded for EEG/EMG in conjunction with a piezoelectric system (Signal Solutions, LLC, Lexington, KY, USA) which uses breathing-related movements to estimate sleep-wake state[76]. Briefly, the piezoelectric platform comprises a polycarbonate cage and a floor covered with a polyvinylidine difluoride (PVDF) film (17.8 cm × 17.8 cm, 110 μm thick; Measurement Specialties, Inc., Hampton, NY) covered with standard rodent litter. A 6-feature vector is extracted from the piezoelectric signal for consecutive 4s-epochs, which was matched to EEG/EMG derived sleep-wake states. Respiratory rate is one of the vector features.

**Frequency-domain analyses of EEG signals.** Power spectral density for each 4 s EEG epoch were generated using a discrete Fourier transform after using a Hamming window, yielding power density spectra (0–100 Hz) with a frequency resolution of 0.25 Hz. Bins containing frequencies between 49 and 51 Hz were excluded due to power-line artifacts in some animals. Epochs containing signal artifacts were separately identified to be included in state quantification though not in spectral analysis. Of note, frequencies are annotated based on midpoint of 0.25 Hz bin (e.g. 2.0 Hz = 1.875–2.125 Hz). δ-power was calculated for 4-s-epochs of NREMS surrounded by artifact-free epochs of the same behavioral state. Consolidated NREMS episodes were defined as uninterrupted sequences of at least 32 s (8 4-s-epochs), based on visual inspection and previous publications[7,28].

**Time-course analysis of spectral power in NREMS.** The spectral band (0.25–90 Hz), was separated into its components to examine dynamics across time of individual bands: δ (1–4 Hz), δ1 (0.75–1.75 Hz), δ2 (2.5–3.5 Hz), theta (6–9 Hz), sigma (10–15 Hz), and beta/low gamma (18–45 Hz). To control for interindividual differences, the absolute spectral power of the each of these frequency bands, was expressed to values calculated to the period during the two baseline days with the lowest mean value, corresponding to the last 4 h of the light period (ZT8–12). Values were averaged into time periods containing an equal number of 4-s-epochs scored as NREMS, referred to as quantiles. For the light periods 12 quantiles were chosen, for the dark periods 6, and in the light period immediately following SD 25.

**Time-course analysis of δ-dynamics at transitions into NREMS.** Following SD, the dynamics of the sub-bands (δ1 and δ2), were analyzed per 4-s-epoch across transitions from either wake or REMS, to NREMS, similar to previously published methods[7]. Transitions were separated into four groups, the first three quantiles calculated starting at SD recovery, and the ensuing hours following the pivot point (see section below). A Wake/REMS-to-NREMS transition was defined as ≥4 consecutive 4-s-epochs of NREMS, preceded by ≥8 scored as either wake or REMS. δ1- and δ2-spectral power was calculated across these transitions starting at 1 min before and 2 min after, and then averaged across these subsequent 3 min windows. Values were expressed relative to changes during two baseline recording days (ZT8-12), as for δ-power. Rise-to-maximum curves for the first quantile were used to quantify the rapid ascent of δ2 as compared to δ1 using SigmaPlot (v. 12.5 SysStat 2011).

**Slow-wave slope analysis.** Raw EEG signals at 200 Hz were imported into Matlab (v. 2019b, Mathworks Inc.), in addition to sleep-wake states. EEG were then filtered for δ-spectrum (Chebyshev type-II 0.5–4.5 Hz, bandstop at 0.1 and 10 Hz, using a zero-phase digital filtering function in Matlab, *filtfilt*), similar to others[16,19,25,77]. To detect slow-waves, a custom-made Matlab algorithm was employed, based on zero-crossings and wave reconstruction closely following others[16,26]. Detection of

NREMS SWs was achieved using the following steps: (1) zero-crossings were determined, (2) surrounding local maxima and minima were identified, (3) thresholding was applied to control for SW-amplitude (see below), and (4) mathematical slope based on SW-amplitude and -frequency (period) was determined in all possible directions and combinations (see Supplementary Fig. 1). The slopes which are analyzed in the manuscript refer to zero-crossing to maximum amplitude SWs which begin with a negative deflection. Though all slopes were calculated (Supplementary Fig. 1), their mean relative changes across time were nearly identical. Data was then grouped and averaged for NREMS episodes >32 s. Slope, amplitude, and period changes were expressed as a percentage of baseline ZT8-12 NREMS values, as with spectral power.

The above algorithm detected SWs across the entirety of the 96 h experiment for all NREMS episodes and lower and upper thresholding was applied to capture bona-fide true NREMS SWs based on visual inspection and previously published criteria[16,25]. Upper thresholds were set at 6 times the s.d. of the amplitude of all detected waveforms. The lower threshold was set at the 95% probability limit of SWs detected during REMS, with the idea that SWs would not appear during REMS. For amplitude distribution, see Supplementary Fig. 1d. Following SW detection, SWs were categorized according to their frequency in 0.25 Hz windows (between 0.75 Hz and 4.5 Hz) and analyzed for incidence (and density min−1) and amplitude (see Fig. 2a, b and Supplementary Fig. 2a, b).

Average waveforms during NREMS episodes (Fig. 2g, Supplementary Fig. 2d) were first located using the filtered EEG signal for δ1 and δ2, and then interpolated to fit 800 ms (δ1) and 300 ms (δ2), to better compare individual SWs of different frequencies within these ranges. The timecodes of these SWs were then reutilized to extract raw unfiltered waveforms. To determine onset and offset of nested δ2-waves, locations of δ1-waveforms were stored and the raw signal was then filtered only for δ2 (2.5–3.5 Hz) using the same Chebyshev filter characteristics (see above). δ2-waveforms which fell within these timecodes were considered nested and their onset/offset from δ1-waveforms was calculated. Of note, although 38 mice were used for frequency-domain (spectral) analysis, one mouse was removed for SW-slope analysis due to the present of intervening artifacts during certain NREMS episodes, which hampered zero-crossing detection and thresholding in this animal.

**Analysis of physiological variables.** All variables presented in Fig. 7 and Supplementary Fig. 6 (with the exception of heart rate), were expressed in relative values and grouped into identical quantiles as δ1 and δ2 (Fig. 5a). In mice this was 25 quantiles (5 prior to pivot-point, and 20 after). Specific analyses are as follows: Higher frequencies (Fig. 7b) are calculated identically to δ-power, except at different frequencies (18–45 Hz). Muscle tone (Fig. 7c) is calculated as the relative changes in mean EMG variance across NREMS episodes. EMG variance is calculated based on the sum of the squared distances of from the mean, divided by the number samples (800). Values are expressed relative to EMG variance during NREMS of the baseline (ZT8-12). Brain temperature (Fig. 7d) is represented as differences during NREMS quantiles from average values during the 6 h waking period (SD) preceding the recovery. Respiration (Fig. 7e) is described in a previous section and is expressed in absolute frequency.

**Hierarchical clustering to determine δ-band separation.** Cluster analysis of referenced (baseline ZT8-12) power per 0.25 Hz, as in Fig. 2d, was achieved based on mean values from 38 mice for the first 25 quantiles of NREMS during SD recovery (see main text). The Matlab function *clustergram* was used to generate a dendrogram and heat map, using hierarchical clustering based on Euclidean distance metric and average linkage.

**Detection of ON-/OFF-states and neuronal firing rates.** Detection of ON-/OFF-states was achieved using bandpass-filtered (0.5–3 Hz) LFP/EEG signals, in forward and reverse directions (*filtfilt*, Matlab). Individual ON-/OFF-states were detected using zero-crossing method of these signals. ON-state onset was defined as crossing from negative to positive, and those ON-/OFF-states with an amplitude <1 s.d. from the means, and shorter than 200 ms, were excluded. To extract multiunit activity from the LFP, signals were bandpass-filtered (600–4000 Hz, fourth-order elliptic filter, 0.1-dB passband ripple, –40 dB stopband attenuation), and detected using a threshold of 7.5x the median of the filtered signal's absolute value. Single units were then identified after sorting with the WaveClus toolbox (v.3). Average firing rate was calculated based on the number of spikes present per second of NREMS episodes and expressed in Hz.

**Human experiments.** A total of 110 healthy young men participated in the four studies used for this analysis. The aggregate results presented were equally observed in each study separately. All human studies were approved by independent Institutional Review Boards and complied with the respective laws and regulations on research in human subjects and the World Medical Association Declaration of Helsinki. No participant had traveled more than two time zones in the preceding 3 months, nor suffered from a diagnosed nervous system disorder or other acute medical condition. Additionally, subjects were pre-screened in the laboratory prior to the study to confirm the absence of any sleep disorders and were free of medication and recreational drug use. All subjects signed informed consent and were compensated financially for participating. For more details see previous publications[32–35]. Some subjects

were removed across four studies due to EEG signal artifacts, which made time-course analysis of spectral bands difficult. For topographical and heart rate analyses, one subject each was removed for signal artifacts.

**Data acquisition**. Continuous recording of EEG, EOG, EMG, and ECG data was acquired during baseline and following a sleep deprivation. Sleep and waking stages were visually scored in 20s-epochs (C3-A2 derivation), using Rembrandt® Analysis Manager (version 8; Embla Systems, Broomfield, CO, USA). Movement- and arousal-related artifacts were visually identified and eliminated from subsequent analysis. Analog signals from each derivation (see Fig. 6c), sampled at 256 Hz were filtered [high-pass (−3 dB at 0.15–0.16 Hz) and low-pass filtering (−3 dB at 67.2 Hz)]. EEG spectra were calculated identically as in mice (see above) using Matlab for each 4s-epoch and averaged according to the assigned sleep-wake state, for each derivation available. NREM/REM-sleep cycles were first defined according to, and for baseline and recovery conditions, δ-power time-course during all scored epochs was visually inspected to separate the first two episodes which did not have a long REMS episode between them. Only the first 8 h of recovery sleep was analyzed. EEG power was expressed to all-night averages of the baseline period. Each NREMS episode was separated into 10 quantiles each containing an equal number of 20-s-epochs scored as NREMS. To compare relative changes across the scalp in a subset of individuals (Fig. 6c, d), mean power across all sites for each δ sub-band during baseline was used. Mean power for electrode site during SD recovery was then estimated at the maximum quantile during for δ1 and δ2 during each NREMS episode and expressed as a log2 ratio to baseline.

Heart rate dynamics during NREMS were expressed as changes in RR-intervals as a percentage of the individual mean baseline NREMS RR-interval (1.07 ± 0.02 s or 57 ± 1 beats min⁻¹). Heart rate variability as an indicator of sympatovagal balance was estimated by determining the LF/HF-ratio; i.e., the ratio between low-frequency (0.04–0.15 Hz) and high-frequency components (0.15–0.4 Hz) of the heart rate spectrum,

**Statistical analyses**. All statistical analyses were performed in either Statistica 8.0 (Statsoft, inc) or using built-in MATLAB functions. Note that p-values from Statistica are limited to 17 decimal places (i.e. $p = 1.0E-17$ is the smallest reported). Sleep-wake distribution, EEG spectral power, and time-course dynamics of specific frequency bands were assessed using two- or three-way repeated-measures analysis of variance (rANOVA). Statistical significance was considered as $p < 0.05$, and all results are given as mean values ± SEM. Tukey's post-hoc test was used to determine significant effects and interactions and corrected for multiple comparison. Comparison of two-groups was achieved with two-sided Student's t-tests. Whenever possible, within-subject paired $t$-tests were used. Statistical methodology is further described in the results section and figure legends. Linear and piece-wise regressions were performed, and Pearson's correlation coefficients were statistically compared by t-test after normalization using the Fisher-Z transformation. Exponential non-linear decays were fitted for Supplementary Fig. 4b, c using the *fitnlm* function in Matlab.

**Reporting summary**. Further information on research design is available in the Nature Research Reporting Summary linked to this article.

## Data availability
The main dataset consisting of 38 male mice implanted for sleep/wake phenotyping and analyzed during the current study are available in the FigShare repository, complete with data descriptors at https://doi.org/10.6084/m9.figshare.12245366. Source data is available as a Source Data file provided with this paper. Other datasets analyzed (optogenetics/tetrode and human EEG experiments) for the current study are available upon request.

## Code availability
Code used to read the datafiles mentioned above are also located at the same FigShare repository, in addition to the slow-wave detection algorithm. Other code is available upon request.

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

## Acknowledgements

We thank Sereina Bodenmann, Kathrin Stingelin, Sebastian C. Holst, and Susanne Weigend for leading the collection and scoring of, and Diego M. Bauer for his assistance in preparing, the human EEG datasets used in this study. We greatly appreciate the insights on this project provided by Peter Achermann, Anita Lühti, Laura Fernandez, and Francesca Siclari. We thank Alexander A. Borbély for critical insights and comments on the bioRxiv pre-print version of this manuscript, as well as suggesting the term wake-inertia. We acknowledge Alpár Lázár and Zsolt Lázár for their initial help with the SW-detection algorithm, and Reto Huber for his discussions about heart rate and sleep in humans. This study was performed at the Universities of Lausanne, Bern, and Zurich, Switzerland, and supported by the Swiss National Science Foundation (SNF no 146694, Sinergia 136201 to P.F.) and the States of Vaud (supporting J.H., M.M.B.H., and P.F.), Bern (supporting T.C.G. and A.R.A.), and the University of Zürich (supporting H.-P.L.). Further support for J.H. was provided by the Dr. Rub Foundation, and the University of Lausanne Foundation, and for A.R.A. by the European Research Council (725850).

## Author contributions

J.H., T.C.G., M.M.B.H., V.M., A.R.A., H.-P.L., and P.F. designed the studies. Experiments were performed by J.H., M.M.B.H., V.M., T.C.G., and Y.E.; J.H. and T.C.G. analyzed the data. J.H., A.R.A., and P.F. wrote the manuscript. All authors commented on and approved the manuscript.

## Competing interests

The authors declare no competing interests.
