## [Peer Review File · Nature Communications]

Reviewers' comments:

Reviewer #1 (Remarks to the Author):

This manuscript provides important novel insights into the relationship between sleep-wake history and EEG dynamics during sleep. As the authors emphasize, the "increase in delta power" after sleep deprivation became synonymous to sleep homeostatic process, which is an oversimplified view and not an accurate representation of how sleep is regulated or the meaning of SWA and its underlying neurophysiological mechanisms. The results presented here are consistent with previous literature, including both animal and human work, and the study also presents new results and re-analysis of existing data, which altogether make a strong case for heterogeneity within the traditional slow wave frequency band. Understanding the neurophysiological mechanisms underlying the EEG dynamics within the slow frequency range will undoubtedly shed new lights on sleep homeostasis.

This study is timely and important, and the data analyses and presentation are superb. However, I have several comments/suggestions for revision, mostly concerning authors' interpretation of their own and others' results.

Major comments

1. The title of the manuscript is somewhat misleading: it sounds as it calls into question the validity of EEG slow-wave dynamics as a measure for sleep homeostasis, while instead it clarifies the relative contribution of different frequencies within the traditional EEG slow-wave frequency, which have, as many studies have shown previously, different dynamics after sleep deprivation. For example, using a modelling approach, it has been shown that the time constants of Process S are greater for slow frequencies in rats: <https://www.ncbi.nlm.nih.gov/pubmed/17683787>. The shift of power towards slower frequencies (shown previously and by Hubbard et al.), is expected to result in a slower decrease in slower frequencies and a faster decrease in the faster part of SWA range. These observations do not raise any issues of "validity" of SWA dynamics as a proxy of sleep homeostatic process, but they help to clarify the underlying mechanisms. The study of Hubbard et al. highlights important limitations of the overall approach of relying on "global" EEG recording and spectral analysis, rather than showing that SWA is not an adequate measure of Process S.

2. The observation of faster delta waves nested within slower, "multipeak" waves is interesting, and consistent with the findings from human studies, that different "peaks" within a multipeak wave could reflect slow waves originating from different sources. Have you analysed the effects of sleep-wake history, such as baseline vs recovery after sleep deprivation, on the phenomenon of "nesting"? One approach to disentangle the contribution of slower and (nested) faster delta waves into the spectral dynamics after SD could be to filter the signal in delta1 band and subtract it from the original signal (and also do the same for the delta2 band), and calculate the effects of sleep deprivation on spectra calculated from the resulting signals (original minus delta1 and original minus delta2).

3. The effects of optogenetic silencing of CMT neurons on LFP spectral power in the cingulate cortex are interesting, but what happens in other brain areas after CMT silencing? Is the selective upregulation of delta2 band in this case site-specific and localised to the cingulate cortex? Do you have EEGs in other cortical areas in this study? Further, you show that the initial sleep after SD is characterised by enhanced delta2 band and increased brain temperature and higher EMG tone. Does optogenetic silencing of CMT induce these other changes in addition to the increase in delta2 band in the cingulate cortex?

4. Line 569: Contrary to the author's interpretation, the change in SW slope has not been interpreted as an independent SW-feature. It has been shown that SW slope reflects how synchronised is the transition between neuronal population ON to population OFF period. If the entire local network transitions swiftly and synchronously into an OFF period, the latter is more likely to be longer, and, in turn, the length of an OFF period has been associated with SW amplitude. Therefore, I would argue that the key underlying phenomenon for the homeostatic changes in EEG or LFP-derived SWA changes is the level of synchrony across the neuronal population (reflected in the slope), and SW amplitude, if anything, is a secondary feature. The

bottomline is that to my opinion it is very difficult to reach strong conclusions about the meaning of specific features of slow waves, or to understand how they relate to each other, using "global" EEG signals only.

5. Generally, I am not convinced that there are two distinct classes of SWs. The shift in spectral power between high and low sleep pressure condition (shown previously and also demonstrated here) instead suggests a possibility that characteristics of SWs (but more importantly the underlying network dynamics) can change across time – perhaps different characteristics changing at different rates. This may result in faster and slower frequencies within the SW frequency band to behave differently, but it does not mean that they are regulated independently or reflect distinct neurophysiological processes. I would interpret the slowing of the spectral peak not as decreased incidence of delta2 waves, but as a change in network dynamics/synchrony, which results in delta2 waves gradually becoming delta1 waves.

6. The section titled "Implications for models on sleep regulation and function" provides a stimulating discussion, but it does not provide a clear solution to the problem of equating SWA dynamics with sleep homeostatic process (which I totally agree with the authors is an important issue in the field). The paper shows elegantly, building on previous work, that there is a heterogeneity within the traditional, and, of course (!) arbitrary frequency band. Different aspects of delta band dynamics may indeed reflect different aspects of the elusive homeostatic process, but subdividing SWs into delta1 and delta2 does not put the problem to rest.

Minor comments

7. Line 723: It is stated "However, this increase in δ_1 after SD in the mouse could relate to aspects other than extended wakefulness, such as the intervention to keep mice awake." Please elaborate. Is there any experimental evidence available for this statement?

8. The choice of terminology for UP/DOWN states is suboptimal as it was derived from filtering EEG/LFP and not from neuronal activity.

Reviewer #2 (Remarks to the Author):

Reassessing the validity of slow-wave dynamics as a proxy for NREM sleep homeostasis

By Hubbard et al.

In this intriguing manuscript, the authors have (re-) analysed EEG data obtained in mice and man, to investigate the changes in occurrence of slow-waves in the NREM sleep EEG, as the authors noted changes in frequencies of delta waves in mice after sleep deprivation. Subsequently the authors were able to distinguish two types of delta waves (1: 0.75-1.75 Hz and 2: 2.5-3.5 Hz), which show two distinct time courses after sleep deprivation. A similar distinction could be made in the human EEG albeit with both frequencies being slower compared to mice. The dynamics of the delta 2 activity paralleled that of other physiological parameters changing in relation to sleep deprivation. Some ways of expressing and some claims in the manuscript, like for instance the way the two process model of sleep regulation is represented in the manuscript, the claim that the authors 'discovered' two types of delta waves, and the way the relationship between amplitude, incidence and slope of slow-waves are related do not do justice to what has been written about this subject in the past and probably need correction. The detailed analysis and the conclusions about the existence of a transient NREM sleep state are of interest and therefore merit more attention. However, in my opinion the conclusions drawn are a bit over the top and the data do not support the claims made.

Major comments

The consequences for the existing models of sleep regulation are not really fleshed out in the manuscript. In addition, it is a bit worrying that the authors seem to revert their attention to 'popular' versions of these models instead of discussing how their data may be a challenge to the models as they are described in the original manuscripts.

On page 36 the authors shortly summarize the two process model and particularly the homeostatic component of this model, but also build a kind of straw man of the model by writing "Its acceptance is so wide-spread that EEG delta power has now become the equivalent to the underlying sleep homeostatic process and any interventions affecting the manifestation of SWs are taken as a proof of altered sleep homeostasis." However, this is not how the model originally was described by the authors. SWA merely reflects the level of the sleep homeostatic process and in the more elaborate model (Achermann Brain Res Bull 1993) the level of SWA is thought to represent the decrease rate of process S instead of process S itself. Next to that, there is a long list of changes in SWA, either induced pharmacologically or pathological (see for an overview Deboer 2015 Curr Topics in Behavioral Neurosci) in which it is very clear that they do not represent or reflect the level of the sleep homeostatic process. Therefore, the way the homeostatic process is represented on page 36 does not do justice to how it was originally conceptualized by the authors of which the model is now being challenged. I agree with the authors that there is a large group in sleep research who live with the described misconception, but this does not mean that we need to strengthen this. This is also true for the last sentence of the discussion (page 38): "...would allow us to refocus attention on the regulation and function of time-spent-in-NREMS and REM sleep, as their rebounds after SD have not been taken into consideration in the two-process model,..." I think this is not true. I agree that REM sleep has been mostly forgotten in the two process model, but particular when it comes to NREM sleep this is nonsense. The amount of NREM sleep in sleep homeostatic processes has been taken into consideration in all the manuscripts where authors not simply analysed changes in SWA, but included the durations of NREM sleep in this analysis by calculating slow-wave energy. Particularly in rodent studies this has been repeatedly applied, also by some of the authors of the present manuscript, to analyse sleep homeostatic responses. When it comes to the well-known 'human' version of the two process model, I agree, again that REM sleep is forgotten, but that in this model it is very clear where the time-spent-in-NREMS is. It is, obviously, the interaction between S and C that determines the sleep time (NREM and REM), but I think the authors know that as well as I do. I want to urge the authors to not criticize the 'popular' version of the two-process model. This straw man has been built before and does not do justice to the model. Please keep it fair and go back to the original publications.

In addition to that, one can ask the question whether the finding that the sleep homeostatic response (in sleep intensity) consists of two types of waves, disqualify the combined activity of these two waves to be a good marker for sleep homeostasis? The model has proven its value and has shown (over the last 3-4 decades) that there is a close relationship between power density in the slow-wave range, sleep homeostatic modelling and predictions about sleep depth and duration. The finding that this homeostatic response in delta power may be shaped by two types of delta waves is an interesting finding, but this does not mean that the model per se is challenged.

Also the claim in the abstract that the authors 'discovered' two (new) types of delta waves is not completely true as they also acknowledge later in the manuscript. Similar frequency ranges have been used to separate delta activity in a fast and slow component in several previous studies. Basically the authors only confirm, what was suggested in the literature for the last two decades, that there exist two distinct NREM sleep delta frequencies, each with its own time course of change after a sleep deprivation (see for instance Huber et al J Neurophys 2000). The existence of these two types of waves in the EEG is supported by electrophysiological data as reviewed by Amzica and Steriade already in 1998 in *Electroencephalogr Clin Neurophysiol* (which is more than 20 years ago).

A similar dis-representation as is done for the two-process model and the two types of delta waves is then presented on page 37 for the way the analysis of slow-wave amplitudes and slopes is done to support the synaptic homeostasis hypothesis. The finding that the amplitude and incidence fully accounts for the change in slow-wave slopes is not very surprising and seems to suggest the authors do not understand that the analysis of slow-wave slopes in the articles analysing the changes in synchronization of firing rate of cortical neurons to explain the changes in shape of the slow-waves (and subsequently in other publication draw conclusion on neuronal connectivity based

on the slope of the waves) is done on waves with the same amplitude. In these papers waves with the same amplitude are compared before and after sleep deprivation and the authors find that the slope of the waves (with the same amplitude) increases after sleep deprivation, and they take that as a support of the notion that neuronal connectivity is changing. The comparison of waves with the same amplitude is very important in this. The latter is completely ignored by the authors in the present manuscript. And it is, in my view, then a bit peculiar to mix waves of different amplitudes and come up with the statement that amplitude (and incidence) completely account for the change in slope. It seems that the authors completely miss the point here.

Minor comments

Page 3 line 74, the first paper modelling sleep homeostasis in the mouse is missing (Huber et al Brain Res 2000).

Delta 2 also more increased in the dark, is that sleep deprivation? The effect here seem to last longer than after sleep deprivation. (fig 4A)

Line 171-172 I am a bit confused here. I don't think the lowest values for the SW period were present in the first hour of recovery (red circles [in fig 1D?]). They seem to be very average compare to the hours following (they do not have the highest values or the lowest values in the graph). Also in figure S2E period does not seem to be affected, as is also confirmed in the legend of the figure. However, in the main text it is claimed repeatedly that period is shorter.

Line 290: "We found that delta1 power was unaffected [by] time spent in SD..."

Reviewer #3 (Remarks to the Author):

Title: Reassessing the validity of slow-wave dynamics as a proxy for NREM sleep homeostasis

Author: Jeffrey Hubbard et al.

The topic of how cortical oscillations can be used to assess sleepiness and "sleep need" following sleep loss (i.e., NREM sleep homeostasis) is one of long-standing interest to specialists in the field of sleep research. Given the topic is clearly important the evaluation of this manuscript rides on whether the findings represent a sufficiently important advance to warrant publication in the high impact journal Nature Communications.

Summary of findings and potential impact: The paper identifies two types of delta waves in mice and concludes that only the faster waves (2.5 to 3.5Hz in mice) respond to sleep loss with increased amplitude during subsequent recovery sleep, whereas the slower wave (0.75 – 1.75 Hz in mice) do not. Studies in humans confirm the presence of 2 similar types of slow waves that respond the same way. The findings indicate that only the faster type of delta waves have contributed to the past findings supporting the the long-held idea that slow wave power (with a freq range that includes both types of the waves the authors describe here) is a valid proxy for NREM sleep homeostasis. The biological substrate supporting unique generation of these 2 types of slow waves is supported by their differential modulation by inhibition of the centromedial thalamus. The association of high delta waves in NREM sleep immediately after sleep deprivation with other physiological measures that are more closely associated with waking (temperature, muscle tone, heart rate, etc.) is an interesting paradoxical observation that bears further investigation although the explanations given for these data are plausible. Overall, the findings could usher in a new era of electrophysiological studies that more precisely explore cortical slow waves as a measure of sleepiness and the homeostatic sleep drive.

Critique: This is an interesting and well conducted study that warrants publication in Nature Communications. The discussion section is particularly well written and easy to follow, clearly conveying the importance of the findings for the field. The discussion section was critical to this reviewer's final positive decision. All the other sections of the manuscript including the figures are

very complex, difficult to read, understand, and to extract the important findings (for a reviewer/reader who does not study electrophysiological measures of sleep homeostasis).

Consequently, rewriting the manuscript for clarity and ease of communication is warranted prior to publication in this high impact journal for a broad audience of specialists in sleep research.

Response to reviewers:

We are grateful to all three reviewers for their time and effort to help improve our manuscript. We have substantially rewritten the Results section to make it more accessible as to Reviewer 3's recommendation. We did so by focusing more on the main findings of the manuscript and discarding the effects of EEG derivation on delta power dynamics and on the relationships between the various measures that define a slow wave. Both analyses, the latter of which was contested by Reviewers 1 and 2, were not critical to the main claims of the manuscript although they did lead to the observation on which the study was based (i.e., apparent shortening of slow wave period after prolonged period of waking). Thanks to a number of excellent suggestions by Reviewer 1 for additional analyses, other parts of the Results section had, however, to be extended and a figure was added grouping the new optogenetics results. The Discussion section now includes the additional references mentioned by Reviewers 1 and 2 and we have put in an effort to rephrase some of the claims that were judged as not fully supported by the data. However, while we believe that the data we present have repercussion for some of the ideas and concepts in the field and which therefore need to be communicated in the Discussion section, an in-depth consideration of all assumptions that contributed to prevailing hypotheses and models, as we interpreted the main comment by Reviewer 2, would be better suited for a review.

Please find below our point-by-point rebuttal to each of the comments made by the three reviewers. We copied the reviewer's comments followed by our reply in **blue** font. We also took this opportunity to update the literature, to correct errors in the writing, and to improve the layout of some of the figures. The changes made can be seen in a 'red-line' copy of the manuscript. We feel that with all these changes the manuscript has considerably improved and hope the reviewers agree.

Reviewer #1:

This manuscript provides important novel insights into the relationship between sleep-wake history and EEG dynamics during sleep. As the authors emphasize, the "increase in delta power" after sleep deprivation became synonymous to sleep homeostatic process, which is an oversimplified view and not an accurate representation of how sleep is regulated or the meaning of SWA and its underlying neurophysiological mechanisms. The results presented here are consistent with previous literature, including both animal and human work, and the study also presents new results and re-analysis of existing data, which altogether make a strong case for heterogeneity within the traditional slow wave frequency band. Understanding the neurophysiological mechanisms underlying the EEG dynamics within the slow frequency range will undoubtedly shed new lights on sleep homeostasis. This study is timely and important, and the data analyses and presentation are superb.

Many thanks for this very positive assessment on both the quality and importance of our work!

Major comments:

1. The title of the manuscript is somewhat misleading: it sounds as it calls into question the validity of EEG slow-wave dynamics as a measure for sleep homeostasis, while instead it clarifies the relative contribution of different frequencies within the traditional EEG slow-wave frequency, which have, as many studies have shown previously, different dynamics after sleep deprivation. For example, using a modelling approach, it has been shown that the time constants

of Process S are greater for slow frequencies in rats: <https://www.ncbi.nlm.nih.gov/pubmed/17683787>. The shift of power towards slower frequencies (shown previously and by Hubbard et al.), is expected to result in a slower decrease in slower frequencies and a faster decrease in the faster part of SWA range. These observations do not raise any issues of “validity” of SWA dynamics as a proxy of sleep homeostatic process, but they help to clarify the underlying mechanisms. The study of Hubbard et al. highlights important limitations of the overall approach of relying on “global” EEG recording and spectral analysis, rather than showing that SWA is not an adequate measure of Process S.

We can agree that the title might have been provocative, and we now have rephrased it to also emphasize our findings concerning the signs of wake inertia evident in the initial part of recovery NREM sleep. We did not intend to mislead the reader but note, also judged by the comments of Reviewer 2, that ‘sleep homeostasis’ is a charged term.

Indeed, a number of studies already reported on the delta band heterogeneity and we have now added the reference mentioned to the other work already cited from these same authors and others. Nevertheless, we believe the results of our analyses support a somewhat different interpretation.

First, although the frequency specific differences in power dynamics could be interpreted or described as a ‘shift in power towards lower frequencies’, our time-domain analyses demonstrate that such shift does not occur but that rather a population of faster slow waves are responsive to prior wake and recovery by increasing their number and amplitude, while slower slow waves remain largely unresponsive. In a new supplementary figure (Suppl. Fig. 3) we show that this bimodality in the distribution of the periods of slow-waves is maintained throughout the 4-day recording and is thus not an exclusive feature of NREM sleep during which sleep pressure is high. The majority of previous studies documenting slow-wave heterogeneity (and which are cited in the manuscript) used FFT analyses to quantify the frequency-dependent dynamics. However, the FFT analyses cannot quantify aspects of individual slow waves, such as their frequency, and how their occurrence is organized in time.

Second, detailed analyses of the power dynamics in the slower delta waves (referred to as δ_1 in the manuscript) show that they do not decrease in humans and even increase in the mouse, during the initial part of recovery NREM sleep, contrary to what a homeostatic sleep-wake driven process would dictate. Moreover, EEG activity in the δ_1 band does not respond quantitatively to prior time-spent-awake in the mouse. Such observations are incompatible with the assumptions of monotonic increasing and decreasing functions during waking and NREM sleep, respectively. While such functions can still be fit to describe power dynamics in the delta frequency range, this seems not to do justice to the underlying data. Of interest is that in a human ‘nap’ study (Dijk *et al.* J Biol Rhythms 1987) it was determined that the wake-dependent increase in the lowest (i.e., 1Hz) delta frequency bin displayed the fastest dynamics among the slow-wave frequencies, while in the rat study referred to by the reviewer, in which a simulation approach was used, the increase rate of the activity in this lowest delta bin was deemed the slowest (Vyazovskiy *et al.* Brain Res Bull 2007). Our observed δ_1 dynamics in the mouse are more consistent with the step-like fast saturating build-up observed in the human study because, although sleep deprivation increased δ_1 activity, it did so to the same extent after 2, 4, and 6h of enforced waking. We have now mentioned both studies in the Discussion section. Furthermore, we statistically substantiated our claim that δ_2 -power decay in recovery sleep does not follow an

exponential function but that, at least within this 6h recording period considered, is described better with two linear functions with an intersecting pivot point. The outcome of this analysis has been added to Supplementary Fig. 4.

2. The observation of faster delta waves nested within slower, “multipeak” waves is interesting, and consistent with the findings from human studies, that different “peaks” within a multipeak wave could reflect slow waves originating from different sources. Have you analysed the effects of sleep-wake history, such as baseline vs recovery after sleep deprivation, on the phenomenon of “nesting“?

Many thanks for these valuable suggestions. We now have analyzed the “nesting” phenomenon in more detail and presented the new results as an additional figure (Supplementary Fig. 4), which now groups the time course analyses of $\delta 1$ and $\delta 2$ waves prevalence (taken from previous Supplementary Fig. 5A,B) with the newly analyzed sleep-wake driven dynamics of nesting. Across the 96h experiment we calculated: 1) Changes in the percentage of all $\delta 2$ waves which nest inside $\delta 1$ waves; 2) the average number of $\delta 2$ waves nested in $\delta 1$ waves which contain them; 3) the percentage of total $\delta 1$ waves which contain 1 or more $\delta 2$ waves. The analyses show that nesting, quantified as the % of $\delta 1$ waves containing 1 or more $\delta 2$ waves and by the average number of δ waves nesting within a $\delta 1$ wave, is more prevalent when “sleep need” is high (i.e., after sleep deprivation) and quickly reverts to lowest baseline levels within the first 6h of recovery sleep. The percentage of all $\delta 2$ waves that are nested follows a different time course and is low immediately after prolonged waking period because the overall prevalence of $\delta 2$ waves greatly increased ($\delta 2$ waves outside $\delta 1$ waves more so than nested $\delta 2$ waves) while the increase in $\delta 1$ -wave prevalence was delayed. We have described these new findings in the Results section and cite the publication referred to by the reviewer.

One approach to disentangle the contribution of slower and (nested) faster delta waves into the spectral dynamics after SD could be to filter the signal in delta1 band and subtract it from the original signal (and also do the same for the delta2 band), and calculate the effects of sleep deprivation on spectra calculated from the resulting signals (original minus delta1 and original minus delta2).

We have followed this suggestion of the reviewer and calculated spectral dynamics of filtered signals. We find that subtracting a signal containing only $\delta 1$ or only $\delta 2$ activity from the original unfiltered signal removes all activity in the two respective frequency ranges (see Rebuttal Fig. 1A,B below). This removal did not affect the dynamics of the unaffected $\delta 2$ sub-band (see insets). Apparently, the FFT captures the activity in $\delta 2$ waves regardless of whether they are nested or not and captures the activity in $\delta 1$ waves regardless of whether they contain a $\delta 2$ wave or not. In a further attempt to disassociate the effects of nested $\delta 2$ waves on $\delta 1$ spectral dynamics, we subtracted the $\delta 2$ waves identified with the time-domain analyses from the raw signal and re-analyzed both the spectral profiles. This was done 1) irrespective of their association with $\delta 1$ waves, 2) according to their occurrence outside of $\delta 1$ waves, or 3) for those nested within $\delta 1$ waves (Rebuttal Fig. 1C, D, and E, respectively). Although this analysis removed spectral power in the $\delta 2$ frequency range as a function of the number of $\delta 2$ waves removed in each of the 3 calculations, it did not affect the $\delta 1$ power time course. The latter finding is consistent with the results of Rebuttal Fig. 1B, in which all EEG activity in the $\delta 2$

frequency range was removed without affecting $\delta 1$ dynamics. Given this, it seems that the FFT is not suited to addressing questions concerning the temporal organization of different wave elements in the EEG signal.

3. The effects of optogenetic silencing of CMT neurons on LFP spectral power in the cingulate cortex are interesting, but what happens in other brain areas after CMT silencing? Is the selective upregulation of delta2 band in this case site-specific and localised to the cingulate cortex? Do you have EEGs in other cortical areas in this study?

The reviewer raises an important point and the results of the proposed analyses strengthened the conclusions of the manuscript. Thank you!

We did record LFP signals from barrel and visual cortices in addition to cingulate cortex during the CMT silencing experiment and have analyzed $\delta 1$ and $\delta 2$ dynamics during recovery sleep also for these signals. Concurrent with our observations in the cingulate, we find that CMT silencing increases $\delta 2$ activity (but not $\delta 1$ activity) in the visual cortex. Interestingly, CMT silencing did not affect $\delta 1$ nor $\delta 2$ dynamics in the barrel cortex. This finding is consistent with our previous work, which demonstrated the barrel cortex to be “out-of-circuit” from higher-order thalamocortical networks during sleep (Gent *et al.* Nat Neurosci 2018). Evidence of this out-of-circuit property was further confirmed with additional new data we provided showing that 10 seconds of CMT silencing during NREM sleep suppresses both $\delta 1$ and $\delta 2$ activity in the cingulate and visual cortex but not in the barrel cortex. This analysis also revealed a rebound in $\delta 2$ but not $\delta 1$ in cingulate and visual cortex demonstrating its frequency specificity also at this level of regulation. We have now included a new main figure dedicated to these new optogenetic data (Fig. 5).

Further, you show that the initial sleep after SD is characterised by enhanced delta2 band and increased brain temperature and higher EMG tone. Does optogenetic silencing of CMT induce these other changes in addition to the increase in delta2 band in the cingulate cortex?

We now have analyzed the EMG dynamics during CMT silencing. We find that changes in EMG tone did not differ from control conditions suggesting that the specific changes in $\delta 2$ activity are not driving the other variables that characterize ‘waking inertia’ in early NREM sleep and *vice versa* as we already demonstrated with modifying brain temperature prior to pivot point. We have mentioned the EMG result in the Results section. Brain temperature was not recorded during the CMT silencing experiment.

4. Line 569: Contrary to the author’s interpretation, the change in SW slope has not been interpreted as an independent SW-feature. It has been shown that SW slope reflects how synchronised is the transition between neuronal population ON to population OFF period. If the entire local network transitions swiftly and synchronously into an OFF period, the latter is more likely to be longer, and, in turn, the length of an OFF period has been associated with SW amplitude. Therefore, I would argue that the key underlying phenomenon for the homeostatic changes in EEG or LFP-derived SWA changes is the level of synchrony across the neuronal population (reflected in the slope), and SW amplitude, if anything, is a secondary feature. The bottomline is that to my opinion it is very difficult to reach strong conclusions about the meaning

of specific features of slow waves, or to understand how they relate to each other, using “global” EEG signals only.

We can agree that our analyses concerning this specific aspect are not conclusive. In addition, Reviewer 2 also expressed issues with our SW-slope analyses and interpretation (see our response to this reviewer below). SW-slope dynamics and how they relate to other SW characteristics were however secondary to the main findings and conclusions of the manuscript and were included mainly because this question motivated the initial stage of this project. We have decided to remove these analyses and statements from an otherwise already analysis-heavy manuscript. Figure 1 has been modified accordingly and the previous Supplementary Fig. 2 has been removed. We also hope that by doing so, the Results section has become more accessible as requested by Reviewer 3.

5. Generally, I am not convinced that there are two distinct classes of SWs. The shift in spectral power between high and low sleep pressure condition (shown previously and also demonstrated here) instead suggests a possibility that characteristics of SWs (but more importantly the underlying network dynamics) can change across time – perhaps different characteristics changing at different rates. This may result in faster and slower frequencies within the SW frequency band to behave differently, but it does not mean that they are regulated independently or reflect distinct neurophysiological processes. I would interpret the slowing of the spectral peak not as decreased incidence of delta2 waves, but as a change in network dynamics/synchrony, which results in delta2 waves gradually becoming delta1 waves.

The scenario of $\delta 2$ waves gradually becoming $\delta 1$ waves, is indeed an alternative hypothesis when analyzing the spectral dynamics. The time-domain analyses, however, revealed a persistent bi-modality throughout the 4-day recording (new Supplementary Fig. 3), indicating the presence of two populations of SWs that do not gradually change in their dominant period as sleep pressure dissipates. In addition, our optogenetic results suggest that the underlying neurophysiology of $\delta 1$ and $\delta 2$ differs. Moreover, genetic analyses further underscore that different (molecular) mechanisms are likely to underlie the two delta components (e.g. Diessler, Jan, *et al.* PLOS Biol 2018). Reviewer 2 agrees with our interpretation and states: “*The existence of these two types of waves in the EEG is supported by electrophysiological data as reviewed by Amzica and Steriade already in 1998 [...].*”

6. The section titled “Implications for models on sleep regulation and function” provides a stimulating discussion, but it does not provide a clear solution to the problem of equating SWA dynamics with sleep homeostatic process (which I totally agree with the authors is an important issue in the field). The paper shows elegantly, building on previous work, that there is a heterogeneity within the traditional, and, of course (!) arbitrary frequency band. Different aspects of delta band dynamics may indeed reflect different aspects of the elusive homeostatic process, but subdividing SWs into delta1 and delta2 does not put the problem to rest.

Thank you for this positive comment. In a landmark paper by Achermann and Borbély (Neurosci 1997), in which the slower SWs (<1 Hz) were found not to decrease from NREM sleep episode 1 to 2 after sleep deprivation, it was concluded that “[...] *it will be important in future studies to analyze the various low-frequency components separately and to take into consideration their*

different responses to challenges of sleep regulation". Several subsequent studies have indeed documented this phenomenon, however we, as a field, have largely ignored this issue. We have more or less applied the frequency range defined in humans to other species without addressing this important question. If $\delta 1$ in humans cannot be taken as a reliable sleep homeostatic proxy, as the 1997 publication suggests, we should also rethink using $\delta 1$ as proxy in other species, bearing in mind that the homologous frequency range might differ. To move forward and to remove some of the arbitrariness the reviewer eluded to, we should redefine the delta band when used as proxy for sleep homeostatic processes to include only those frequencies in which dynamics follow a "homeostatic" time course; i.e., are in a predictive and quantitative relationship with prior time spent awake and asleep. Although it doesn't put the issue at rest, by narrowing down the frequency range best gauging prior wakefulness, as well as narrowing down the time required to relax back to baseline it puts constraints on the possible neurophysiological mechanisms contributing to the phenomenon. The involvement of thalamic structures, such as the CMT, can be seen as a first step in that direction. We have made this point clearer in the discussion.

Minor comments:

7. Line 723: It is stated "However, this increase in $\delta 1$ after SD in the mouse could relate to aspects other than extended wakefulness, such as the intervention to keep mice awake." Please elaborate. Is there any experimental evidence available for this statement?

Thank you for catching this incomplete line of thought. We have remedied this. Initially, we were thinking of sleep-deprivation associated stress which, quantified as circulating corticosterone levels, does not monotonically increase over the course of a sleep deprivation and already peaks after 3h (Raven *et al.* J Sleep Res 2019). Stress, in a different experimental paradigm (social defeat) has been shown to cause pronounced increases in EEG delta power (e.g. Meerlo *et al.* Neurosci Lett 1997). These EEG changes seem, however, not specific to $\delta 1$ (Fujii *et al.* Front Neurosci 2019). Observations that are likely to be more relevant are that the sleep-deprivation associated increase in $\delta 1$ activity depends on cortical noradrenalin (NA) levels (Cirelli *et al.* J Neurosci 2005) and that these levels are up with sleep deprivation but do monotonically rise with time spent awake and, especially in frontal cortical area, already peak after 3h (Belleesi *et al.* Sleep 2016). The increase in $\delta 1$ after 2h of sleep deprivation that did not further increase with 4- and 6h sleep deprivation might relate to this phenomenon.

8. The choice of terminology for UP/DOWN states is suboptimal as it was derived from filtering EEG/LFP and not from neuronal activity.

Correct, we now refer to the LFP defined UP/DOWN states as ON/OFF states in both the text and the figures.

Reviewer #2:

In this intriguing manuscript, the authors have (re-) analysed EEG data obtained in mice and man, to investigate the changes in occurrence of slow-waves in the NREM sleep EEG, as the authors noted changes in frequencies of delta waves in mice after sleep deprivation. Subsequently the authors were able to distinguish two types of delta waves (1: 0.75-1.75 Hz and 2: 2.5-3.5 Hz), which show two distinct time courses after sleep deprivation. A similar distinction could be made in the human EEG albeit with both frequencies being slower compared to mice. The

dynamics of the delta 2 activity paralleled that of other physiological parameters changing in relation to sleep deprivation. Some ways of expressing and some claims in the manuscript, like for instance the way the two process model of sleep regulation is represented in the manuscript, the claim that the authors ‘discovered’ two types of delta waves, and the way the relationship between amplitude, incidence and slope of slow-waves are related do not justice to what has been written about this subject in the past and probably need correction. The detailed analysis and the conclusions about the existence of a transient NREM sleep state are of interest and therefore merit more attention. However, in my opinion the conclusions drawn are a bit over the top and the data do not support the claims made.

Thank you for the positive comments as well as for the critique, with which we have to disagree with the exception perhaps of the SW-slope analyses that might be regarded as premature (see below and comment to Reviewer 1). We hope to clarify some of the misunderstandings below.

Major comments:

The consequences for the existing models of sleep regulation are not really fleshed out in the manuscript. In addition, it is a bit worrying that the authors seem to revert their attention to ‘popular’ versions of these models instead of discussing how their data may be a challenge to the models as they are described in the original manuscripts. On page 36 the authors shortly summarize the two process model and particularly the homeostatic component of this model, but also build a kind of straw man of the model by writing “Its acceptance is so wide-spread that EEG delta power has now become the equivalent to the underlying sleep homeostatic process and any interventions affecting the manifestation of SWs are taken as a proof of altered sleep homeostasis.” However, this is not how the model originally was described by the authors. SWA merely reflects the level of the sleep homeostatic process and in the more elaborate model (Achermann Brain Res Bull 1993) the level of SWA is thought to represent the decrease rate of process S instead of process S itself. Next to that, there is a long list of changes in SWA, either induced pharmacologically or pathological (see for an overview Deboer 2015 Curr Topics in Behavioral Neurosci) in which it is very clear that they do not represent or reflect the level of the sleep homeostatic process. Therefore, the way the homeostatic process is represented on page 36 does not do justice to how it was originally conceptualized by the authors of which the model is now being challenged. I agree with the authors that there is a large group in sleep research who live with the described misconception, but this does not mean that we need to strengthen this.

The above quoted statement was clearly meant as criticism to current practice of many in the field. We did not think it could be misunderstood to mean that we support this thinking. Reviewer 1 and, judged by his/her last sentence above, this reviewer also seems to understand this statement as criticism and agrees with it. By no means do we support these ‘popular’ versions. We are now more explicit in this criticism and cite the review the reviewer mentioned.

We did not intend, and our results do not challenge, the basic assumptions of the two-process model, although the reviewer seems to be of that opinion. We do, however, call into question the practice of lumping all frequencies into a single frequency band while knowing that only some follow dynamics that are compatible with the assumptions of Process S in the two-process model. The results of our study do however indicate that dynamics of Process S are faster as was assumed using the entire delta frequency band and thus recovery of homeostatic sleep pressure is achieved faster.

This is also true for the last sentence of the discussion (page 38): "...would allow us to refocus attention on the regulation and function of time-spent-in-NREMS and REM sleep, as their rebounds after SD have not been taken into consideration in the two-process model,..." I think this is not true. I agree that REM sleep has been mostly forgotten in the two process model, but particular when it comes to NREM sleep this is nonsense. The amount of NREM sleep in sleep homeostatic processes has been taken into consideration in all the manuscripts where authors not simply analysed changes in SWA, but included the durations of NREM sleep in this analysis by calculating slow-wave energy. Particularly in rodent studies this has been repeatedly applied, also by some of the authors of the present manuscript, to analyse sleep homeostatic responses. When it comes to the well-known 'human' version of the two process model, I agree, again that REM sleep is forgotten, but that in this model it is very clear where the time-spent-in-NREMS is. It is, obviously, the interaction between S and C that determines the sleep time (NREM and REM), but I think the authors know that as well as I do. I want to urge the authors to not criticize the 'popular' version of the two-process model. This straw man has been built before and does not do justice to the model. Please keep it fair and go back to the original publications.

We disagree and although a rebuttal is hardly a suitable platform to discuss these matters in a fair and constructive manner (the stakes in this are highly asymmetric), below some thoughts on why we think that the recovery dynamics of time spent in NREM sleep are not accounted for in the two process model. First, although indeed used by many as an additional metric, in the original two-process model publications (Borbély, *Human Neurobiol* 1982; Daan *et al. Am J Physiol* 1984), the recent reappraisal of this model (Borbély *et al. J Sleep Res* 2016), as well as in the more elaborate model the reviewer refers to above (Achermann *et al. Brain Res Bull* 1993) there is no mention of 'slow-wave energy' nor on the homeostatic processes regulating time spent in NREM sleep. Although the model predicts sleep duration in a circadian context (i.e. it depends on the sleep onset time), the model's emphasis is clearly on the remarkably precise regulation of the 'intensity' dimension of NREM sleep.

While it is abundantly clear that EEG delta power is driven by the sleep-wake distribution, there is hardly any proof that EEG delta power (or Process S) drives the extra time spent in NREM sleep (as compared to baseline levels) during recovery from sleep deprivation. For instance, the model cannot explain why rats sleep more when sleep propensity (EEG delta power) is maintained at values below baseline (i.e., the "negative rebound"; Franken *et al. Am J Physiol* 1991). Clearly, EEG delta power and time spent in NREM sleep are regulated by different mechanisms, which can be deduced from their very different time constants of the respective recovery patterns, their underlying neurophysiology, and even genetics. Combining these two variables into one (i.e., slow wave energy), impedes the assessment of the neurophysiological underpinnings of each contributing process, and would require the formulation of yet another process that keeps track both of Process S and of NREM sleep time. Moreover, leading hypotheses on sleep function, such as the synaptic homeostasis hypothesis (SHY) or the notion that sleep is local and use-dependent, highlight EEG delta power as the regulated feature and not time spent in NREM sleep. We assume that we will not have convinced the reviewer by any of the above arguments but clearly there exist other opinions on this matter.

In addition to that, one can ask the question whether the finding that the sleep homeostatic response (in sleep intensity) consists of two types of waves, disqualify the combined activity of

these two waves to be a good marker for sleep homeostasis? The model has proven its value and has shown (over the last 3-4 decades) that there is a close relationship between power density in the slow-wave range, sleep homeostatic modelling and predictions about sleep depth and duration. The finding that this homeostatic response in delta power may be shaped by two types of delta waves is an interesting finding, but this does not mean that the model per se is challenged.

The modeling approaches the reviewer refers to are based on the premise that EEG delta power, or changes therein, reflect the sleep homeostat. At the same time, the dynamics of the sleep homeostat (Process S) are based on the dynamics of EEG delta power, which turns this into a circular argument. But what if some types of slow waves better reflect this process? Or worse, what if activity in some delta frequencies power further increase during periods when recovery sleep is thought to be most intense? In a publication mentioned by Reviewer 1 (Vyazovskiy *et al.* Brain Res Bull 2007), results of mathematical modelling find that the time constants for Process S' build-up rate can range from ca. 8 to 25h depending on the delta frequency bin chosen. It is hard to imagine that one and the same underlying process can be reflected by such large variations in these dynamics. Nowhere in our manuscript do we challenge the assumptions of the two-process model, but by narrowing down the relevant frequencies, we refine this model and try to get a better grip on the underlying circuitry, hence our optogenetics experiments.

Also the claim in the abstract that the authors 'discovered' two (new) types of delta waves is not completely true as they also acknowledge later in the manuscript. Similar frequency ranges have been used to separate delta activity in a fast and slow component in several previous studies. Basically the authors only confirm, what was suggested in the literature for the last two decades, that there exist two distinct NREM sleep delta frequencies, each with its own time course of change after a sleep deprivation (see for instance Huber et al J Neurophys 2000). The existence of these two types of waves in the EEG is supported by electrophysiological data as reviewed by Amzica and Steriade already in 1998 in *Electroencephalogr Clin Neurophysiol* (which is more than 20 years ago).

We removed 'discovered' from the Abstract but nevertheless insist that the time-domain analyses brought to light an aspect that has not been previously captured using frequency-domain analyses. The latter type of analysis cannot differentiate between the amplitude, slope, and incidence of different types of slow waves contributing to the signal nor can it give insight into the temporal organization of different wave elements in the EEG signal. The identification of two spectral bands with different dynamics cannot differentiate whether the initial selective high activity of faster slow-waves and its subsequent decline is due to slow waves gradually slowing in frequency (i.e., sampling from one population as Reviewer 1 favors) or whether there are two populations each with different response kinetics, which is the scenario the time-domain analysis favors. We had already cited the work of Huber et al. for the reasons mentioned by the reviewer.

Also the work of Amzica and Steriade was already cited to discuss the issue raised by the reviewer, i.e., that different types of slow waves have been described that might relate to the bimodal distribution we identified. However, as to which of the various types of slow waves (slow oscillations, delta waves of cortical origin, and thalamic clock-like delta oscillations) contribute to the slow waves captured at the level of the EEG, and which among them underlie the two slow-wave populations we describe, is unclear. Moreover, the prevalence of the

respective slow waves mentioned in that review were not evaluated in a sleep homeostatic context. We therefore do not share the reviewer's confidence that the two populations we identified correspond to any of the types of slow waves the brain is able to produce depending on brain structure and experimental condition (*in vitro*, *in vivo*, under anesthesia etc). We had already addressed these issues in the discussion where we proposed that δ_1 is not a delta oscillation but the murine homologue of the slow oscillation, while δ_2 reflects the delta frequency range.

A similar dis-representation as is done for the two-process model and the two types of delta waves is then presented on page 37 for the way the analysis of slow-wave amplitudes and slopes is done to support the synaptic homeostasis hypothesis. The finding that the amplitude and incidence fully accounts for the change in slow-wave slopes is not very surprising and seems to suggest the authors do not understand that the analysis of slow-wave slopes in the articles analysing the changes in synchronization of firing rate of cortical neurons to explain the changes in shape of the slow-waves (and subsequently in other publication draw conclusion on neuronal connectivity based on the slope of the waves) is done on waves with the same amplitude. In these papers waves with the same amplitude are compared before and after sleep deprivation and the authors find that the slope of the waves (with the same amplitude) increases after sleep deprivation, and they take that as a support of the notion that neuronal connectivity is changing. The comparison of waves with the same amplitude is very important in this. The latter is completely ignored by the authors in the present manuscript. And it is, in my view, then a bit peculiar to mix waves of different amplitudes and come up with the statement that amplitude (and incidence) completely account for the change in slope. It seems that the authors completely miss the point here.

We agree with this reviewer and with Reviewer 1 that the presentation of these results was not well thought through. Our apologies. For this reason and because of Reviewer 3's recommendation to make the Results section more accessible and because the SW slope/amplitude-analyses are not central to the manuscript, we decided to remove these analyses from the Results section. Figure 1 and the Discussion have been updated accordingly, and Supplementary Figure 2, removed.

Prompted by the comments of the reviewer we wondered whether our findings concerning the higher prevalence of δ_2 waves after sleep deprivation could have any bearing on the increase in slope others reported during this period because faster slow-waves have, on average, steeper slopes compared to slower waves of the same amplitude. We therefore binned amplitude following the approach taken in a recent publication on this topic (Panagiotou *et al.* Sci Rep 2017). We could reproduce that SW slope in early recovery sleep is indeed steeper than in late recovery sleep but also observed a strong correlation ($r = \text{ca. } 0.8$) between frequency and slope within each amplitude bin (see Rebuttal Figure 2 below). By repeating these analyses only on SWs in the δ_2 frequency range, we lost all increases in slope after sleep deprivation. Thus, the increased prevalence of δ_2 waves specifically after sleep deprivation is a plausible and parsimonious alternative explanation to the interpretation that increased SW slopes reflect increased network synchronization and connectivity. We agree with Reviewer 1's statement that "[...] it is very difficult to reach strong conclusions about the meaning of specific features of slow waves, or to understand how they relate to each other, using 'global' EEG signals only."

Minor comments

Page 3 line 74, the first paper modelling sleep homeostasis in the mouse is missing (Huber et al Brain Res 2000).

Reference added.

Delta 2 also more increased in the dark, is that sleep deprivation? The effect here seem to last longer than after sleep deprivation. (fig 4A)

Under baseline conditions mice exhibit sustained episodes of spontaneous waking after dark onset which regularly continue for more than one hour. Delta dynamics in the initial NREM sleep episodes subsequent to these sustained periods of wakefulness show a similar albeit less pronounced differentiation between $\delta 1$ and $\delta 2$ dynamics, which we have illustrated in Figure 3D. $\delta 2$ decay is slower in the dark period as NREM sleep during this time is less prevalent than during the first 6h of the recovery light phase following sleep deprivation. Thus, what is observed is the net effect of waking during which $\delta 2$ increases and NREM sleep during which $\delta 2$ decreases.

Line 171-172 I am a bit confused here. I don't think the lowest values for the SW period were present in the first hour of recovery (red circles [in fig 1D?]). They seem to be very average compare to the hours following (they do not have the highest values or the lowest values in the graph). Also in figure S2E period does not seem to be affected, as is also confirmed in the legend of the figure. However, in the main text it is claimed repeatedly that period is shorter.

Indeed confusing. The red circles referred to results over the 1st three hours of recovery. Because overall SW-period changes quickly, this time interval included data points for which SW period was again close to average. As we have removed the SW-slope/amplitude correlation analyses from the manuscript, we also removed these panels from Figure 1.

Line 290: "We found that delta1 power was unaffected [by] time spent in SD..."

Corrected.

Reviewer #3:

Critique: This is an interesting and well conducted study that warrants publication in Nature Communications. The discussion section is particularly well written and easy to follow, clearly conveying the importance of the findings for the field. The discussion section was critical to this reviewer's final positive decision. All the other sections of the manuscript including the figures are very complex, difficult to read, understand, and to extract the important findings (for a reviewer/reader who does not study electrophysiological measures of sleep homeostasis).

Consequently, rewriting the manuscript for clarity and ease of communication is warranted prior to publication in this high impact journal for a broad audience of specialists in sleep research.

Many thanks for your positive evaluation of our work and to recommend publication in Nature Communications. We can agree that the Results section is dense and difficult to follow. Part of the analyses dealt with how we arrived at the phenotype (shortening of SW period) that led into the main part of manuscript. We have shortened this first part by removing the issues concerning SW slopes versus SW amplitudes and effects of electrode position on delta power dynamics that are not pertinent to the paper's main message. The statistics that interrupted the flow were moved to the Figure legend where appropriate. We have put in an effort to enhance readability throughout the manuscript. We have provided three versions of the revised manuscript; besides the new, 'clean' version, we provided two 'redline' copies: one with all changes made marked as track-changes and one highlighting only the main additions and deletions for readability.

Rebuttal Figure 1: NREM sleep EEG spectra and delta power dynamics of filtered EEGs. Example of average EEG spectra in one animal during the first 6h following sleep deprivation. **(A)** Original unfiltered (raw, black) and ‘ $\delta 1$ -subtracted’ (blue) spectra. **Insert:** time-course of $\delta 2$ power across the 6h based on the raw vs. the subtracted signal. Note complete overlap of the two. **(B)** Original unfiltered (raw, black) and ‘ $\delta 2$ -subtracted’ (orange) spectra. **Insert:** time-course of $\delta 1$ power across the 6h based on the raw vs. the subtracted signal. Note complete overlap of the two. **(C)** As in B but instead of subtracting all $\delta 2$ activity, all detected $\delta 2$ -waves outside $\delta 1$ waves were subtracted from the original signal before spectral analyses. **Insert:** time-course of $\delta 1$ (blue) and $\delta 2$ (orange) across the 6h. **(D)** As in C but for all detected $\delta 2$ slow-waves. **(E)** As in C but for detected $\delta 2$ slow-waves within $\delta 1$ waves. Note that in C-D inserts $\delta 1$ dynamics is unaffected by subtracting $\delta 2$ waves. For all inserted panels delta power expressed as % of the ‘raw’ power obtained in the last interval of recovery in either $\delta 1$ (0.75-1.75Hz) or $\delta 2$ (2.5-3.5Hz).

Rebuttal Figure 2: Changes in SW-slope with amplitude binning in ‘early’ and ‘late’ recovery NREM sleep. (A) SW-slope following amplitude binning (quintiles of 20 percentiles) is higher in all amplitude bins in early (left; 0-2h after sleep deprivation) vs. late (right data point; 2-6h) recovery NREM sleep consistent with published data. (B) SW-slope to SW-frequency correlations during the 6h recovery show high correlations, regardless of amplitude bin. (C) When corrected for frequency, by re-filtering raw signals for the $\delta 2$ band (2.0-4.5Hz, as in **Fig. 2 H-I**), changes in SW-slope during recovery disappear. Analyses based on 37 C57BL/6J mice.

REVIEWERS' COMMENTS:

Reviewer #1 (Remarks to the Author):

I do not have any further comments on the manuscript.

Reviewer #2 (Remarks to the Author):

Rapid decay in fast delta following prolonged wakefulness marks a phase of wake-inertia in NREM sleep

By Hubbard et al.

My first comment on the sentence "Its acceptance is so wide-spread that EEG δ power has now become equivalent to the underlying sleep homeostatic process and any interventions affecting the manifestation of SWs are taken as proof of altered sleep homeostasis." is welcomed with the response that this was clearly meant as criticism to current practice of how many in the field represent the two process model. I am sorry, but I apparently missed the point there, and I am afraid those that are already misrepresenting the model will find confirmation in this sentence instead of criticism. I think criticism on the misconceptions of the popular version should be expressed clearer than that. The addition of an extra sentence "However, it should be obvious that many such interventions can dissociate SW expression from the underlying sleep homeostatic process and must therefore be interpreted with caution", in my mind, does not help, as it does not make the previous sentence clearer. My suggestion is to take out the last two sentences (line 695 to 700) to avoid feeding the misconception. The sentences do not add to the argument the authors want to make, anyway.

The second point is about the time-spend-in-NREMS and REM sleep not being considered in the two-process model. Basically I agree with the authors, and I see the point of the authors, but do think that the issue was presented very black-and-white in the previous version of the manuscript. And that is probably because also this manuscript this is not the place to start a review/discussion on this subject. In my opinion, in the original model (Borbely 1982. Daan et al 1984), which mainly talks about human sleep wake regulation, it is very clear where the extra sleep goes, humans are monophasic sleepers and they increase time-spend in sleep by extending rest, or introducing a nap during the active phase. It is therefore not surprising that the more elaborate model for humans (Acherman et al 1993) mainly focusses on SWA in NREM sleep. And I agree, more should be done to investigate the time-spend in sleep axis. The problem starts in rodent sleep, which is more polyphasic. For rodents it is possible to sleep more (or less) without changing the time spend in rest. The change made in the last sentence is in my view a good solution. Thank you.

Thank you for removing 'discovered' from the abstract, and for changing the argument about the slopes and amplitudes of the slow-waves.

All minor comments were taken care of.

Reviewer #3 (Remarks to the Author):

Unfortunately, a careful reading the critiques of R1 and R2 along with the authors' responses has reduced enthusiasm for the publication of this manuscript in Nature Communications, a high

impact multidisciplinary journal that publishes research in a broad range of topics. In the authors' own words, "We disagree and although a rebuttal is hardly a suitable platform to discuss these matters in a fair and constructive manner.....". The authors do not take their own advice and their lengthy responses to R1 & R2 are seldom helpful. More importantly, the authors' responses to R1 and R2 consistently fail to describe the specific changes they have made to the manuscript to address the reviewer's concerns (and where). Similarly, they fail to indicate when/if changes to the manuscript have NOT been made in response to the reviewer's comments. For example, are the data in the two "rebuttal figures" incorporated into the revised manuscript and, if so, how and where? There are many other instances where the authors are not stating how the reviewer's concerns have been addressed in the revised manuscript.

Minor points:

R1 Q2. The authors response to the response to R1 Q2 was impossible to follow because they refer to Suppl Fig 4 when they meant Suppl Fig 5.

R1 Q3. The authors response here seems contradictory since they say: "Concurrent with our observations in the cingulate, we find that CMT silencing increases $\delta 2$ activity (but not $\delta 1$ activity) in the visual cortex. Interestingly, CMT silencing did not affect $\delta 1$ nor $\delta 2$ dynamics in the barrel cortex." And, in the next sentences later they say, "Evidence of this out-of-circuit property was further confirmed with additional new data we provided showing that 10 seconds of CMT silencing during NREM sleep suppresses both $\delta 1$ and $\delta 2$ activity in the cingulate and visual cortex but not in the barrel cortex."

The term "wake-inertia" is now included in the title of the manuscript, but is not adequately defined.

Reviewer #1 (Remarks to the Author):

I do not have any further comments on the manuscript.

Thank you once more for your numerous comments and suggestions that helped improve the manuscript.

Reviewer #2 (Remarks to the Author):

My first comment on the sentence “Its acceptance is so wide-spread that EEG δ power has now become equivalent to the underlying sleep homeostatic process and any interventions affecting the manifestation of SWs are taken as proof of altered sleep homeostasis.” is welcomed with the response that this was clearly meant as criticism to current practice of how many in the field represent the two process model. I am sorry, but I apparently missed the point there, and I am afraid those that are already misrepresenting the model will find confirmation in this sentence instead of criticism. I think criticism on the misconceptions of the popular version should be expressed clearer than that. The addition of an extra sentence “However, it should be obvious that many such interventions can dissociate SW expression from the underlying sleep homeostatic process and must therefore be interpreted with caution” , in my mind, does not help, as it does not make the previous sentence clearer. **My suggestion is to take out the last two sentences (line 695 to 700) to avoid feeding the misconception. The sentences do not add to the argument the authors want to make, anyway.**

We now have removed the two sentences concerned.

The second point is about the time-spend-in-NREMS and REM sleep not being considered in the two-process model. Basically I agree with the authors, and I see the point of the authors, but do think that the issue was presented very black-and-white in the previous version of the manuscript. And that is probably because also this manuscript this is not the place to start a review/discussion on this subject. In my opinion, in the original model (Borbely 1982. Daan et al 1984), which mainly talks about human sleep wake regulation, it is very clear where the extra sleep goes, humans are monophasic sleepers and they increase time-spend in sleep by extending rest, or introducing a nap during the active phase. It is therefore not surprising that the more elaborate model for humans (Acherman et al 1993) mainly focusses on SWA in NREM sleep. And I agree, more should be done to investigate the time-spend in sleep axis. The problem starts in rodent sleep, which is more polyphasic. For rodents it is possible to sleep more (or less) without changing the time spend in rest. The change made in the last sentence is in my view a good solution. Thank you.

Thank you for removing ‘discovered’ from the abstract, and for changing the argument about the slopes and amplitudes of the slow-waves.

You are more than welcome and many thanks for the critical feedback.

Reviewer #3 (Remarks to the Author):

Unfortunately, a careful reading the critiques of R1 and R2 along with the authors' responses has reduced enthusiasm for the publication of this manuscript in *Nature Communications*, a high impact multidisciplinary journal that publishes research in a broad range of topics. In the authors' own words, "We disagree and although a rebuttal is hardly a suitable platform to discuss these matters in a fair and constructive manner.....". The authors do not take their own advice and their lengthy responses to R1 & R2 are seldom helpful.

More importantly, the authors' responses to R1 and R2 consistently fail to describe the specific changes they have made to the manuscript to address the reviewer's concerns (and where). Similarly, they fail to indicate when/if changes to the manuscript have NOT been made in response to the reviewer's comments. For example, are the data in the two "rebuttal figures" incorporated into the revised manuscript and, if so, how and where? There are many other instances where the authors are not stating how the reviewer's concerns have been addressed in the revised manuscript.

We are sorry to learn that this Reviewer did not find our rebuttal to the criticisms of the other two reviewers instructive. However, given that both these reviewers seemed satisfied with our response and with the many changes we made in the revised version, we assume at least they found it instructive. The length of the rebuttal seemed adequate given the many and lengthy suggestions and in depth criticism provided by these two reviewers.

We do not understand how the Reviewer's initial positive assessment on the manuscript's suitability for publication in *Nature Communications* changed because of how we handled our response to the other two reviewers. In contrast to this Reviewer, we feel the manuscript has been substantially improved over the initial version because of the many excellent suggestions made by all 3 reviewers.

We did specify whether a change was made in the manuscript in response to a comment. It is however true that we did not indicate the specific line concerned. Again, the reviewers, to which we addressed these comments, did not seem to find this problematic and Reviewer 2 did thank us for the specific changes made. Concerning the figures, all through the rebuttal we indicated when the new analyses we performed led to a new or altered (supplementary) figure. The "Rebuttal figures" were included in the rebuttal only (hence their name) and not in the manuscript. Although we assumed this to be clear, we apologize if this led to a misunderstanding. Of note, following the journal's policy not to refer to '*data not shown*' and since we believe it is an important finding, we now have added Rebuttal Fig. 2 as Suppl. Fig. 8 and replaced the '(not shown)' statement in the Discussion with a reference to the figure.

Minor points:

R1 Q2. The authors response to the response to R1 Q2 was impossible to follow because they refer to Suppl Fig 4 when they meant Suppl Fig 5.

Our apologies for this mix-up.

R1 Q3. The authors response here seems contradictory since they say: “Concurrent with our observations in the cingulate, we find that CMT silencing increases $\delta 2$ activity (but not $\delta 1$ activity) in the visual cortex. Interestingly, CMT silencing did not affect $\delta 1$ nor $\delta 2$ dynamics in the barrel cortex.” And, in the next sentences later they say, “Evidence of this out-of-circuit property was further confirmed with additional new data we provided showing that 10 seconds of CMT silencing during NREM sleep suppresses both $\delta 1$ and $\delta 2$ activity in the cingulate and visual cortex but not in the barrel cortex.”

Not contradictory, but when only reading the rebuttal this is admittedly potentially confusing. The first statement concerned the rebound phenomenon; i.e., during the NREM sleep following CMT silencing, $\delta 2$ -power was boosted. The second statement concerned our new data in which we carefully analyzed how NREM sleep $\delta 2$ -power is affected during CMT silencing. The results were surprisingly congruent; only when during CMT silencing ongoing $\delta 2$ -activity is suppressed in a given cortical domain, does $\delta 2$ -power rebound in subsequent NREM sleep. In the manuscript this was, however, clearly explained (e.g. in the Results: “*Both the suppression and following rebound were specific for the cingulate and visual cortices and not observed in the barrel (Fig. 5C)*” and in the Discussion: “*Whether the $\delta 2$ augmentation is of thalamic or cortical origin is unclear as we quantified SWs during NREMS episodes consecutive to those during which the CMT was optogenetically silenced, indicating that increases in $\delta 2$ -activity might be a cortical rebound phenomenon as SWs were suppressed during preceding optogenetic silencing.*”).

The term “wake-inertia” is now included in the title of the manuscript, but is not adequately defined.

We had described the term wake-inertia in the Discussion as follows: “*We interpret this highly dynamic NREMS phase to reflect a phase of wake inertia during which the system adjusts from the aftereffects of a highly active and sustained waking period before typical NREMS can be reinstated*”. We think this description adequately explains what we mean with the term. Although, we did not want to formally define this phenomenon up front, as it is a new concept better suitable for the Discussion, we now mention wake-inertia also in the before-last sentence of the Introduction: “*This study identified a previously unknown complexity of the central and peripheral processes associated with the aftereffects of prolonged waking, which we refer to as wake-inertia, exemplified by the short-lived decay of a specific sub-population of SWs, before reverting to levels typical of NREMS.*”